# PRIMO: Private Regression in Multiple Outcomes

**Seth Neel**                                                                                      *sneel@hbs.edu*
*Harvard Business School*

**Reviewed on OpenReview:** *https://openreview.net/forum?id=34vtRA3Nvu*

## Abstract

We introduce a new private regression setting we call *Private Regression in Multiple Outcomes* (PRIMO), inspired by the common situation where a data analyst wants to perform a set of $l$ regressions while preserving privacy, where the features $X$ are shared across all $l$ regressions, and each regression $i \in [l]$ has a different vector of outcomes $y_i$. Naively applying existing private linear regression techniques $l$ times leads to a $\sqrt{l}$ multiplicative increase in error over the standard linear regression setting. We apply a variety of techniques including sufficient statistics perturbation (SSP) and geometric projection-based methods to develop scalable algorithms that outperform this baseline across a range of parameter regimes. In particular, we obtain *no dependence on $l$* in the asymptotic error when $l$ is sufficiently large. Empirically, on the task of genomic risk prediction with multiple phenotypes we find that even for values of $l$ far smaller than the theory would predict, our projection-based method improves the accuracy relative to the variant that doesn't use the projection.

## 1 Introduction

Linear regression is one of the most fundamental statistical tools used across the applied sciences, for both inference and prediction. In genetics, polygenic risk scores Krapohl et al. (2018); Pattee & Pan (2020) are computed by regressing different phenotypes (observed outcomes of interest like the presence of a trait or disease) onto individual genomic data (SNPs) in order to identify genetic risk factors. In the social sciences, observed societal outcomes like income or marital status might be regressed on a fixed set of demographic features Agresti & Barbara (2009). In many of these cases where the data records correspond to individuals, there are two aspects of the problem setting that co-occur:

**Aspect 1.** The individuals may have a legal or moral right to privacy that has the potential to be compromised by their participation in a study.

**Aspect 2.** Multiple regressions will be ran using the same set of individual characteristics across each regression with different outcomes, either within the same study or across many different studies.

Aspect 1 has been established as a legitimate concern through both theoretical and applied work. The seminal paper of Homer et al. (2008) showed that the presence of an individual in a genomic dataset could be identified given simple summary statistics about the dataset, leading to widespread concern over the sharing of the results of genomic analyses. In the machine learning setting, where what is being released is a model $w$ trained on the underlying data, there is a long line of research into "Membership Inference Attacks" Hu et al. (2021); Shokri et al. (2016), which given access to $w$ are able to identify which points are in the training set. Over the last decade, differential privacy Dwork & Roth (2014) has emerged as a rigorous solution to the privacy risk posed by Aspect 1. In the particular case of linear regression the problem of how to privately compute the optimal regressor has been studied in great detail, which we summarise in Subsection 2.1.

Aspect 2 has been studied extensively from the orthogonal perspective of multiple hypothesis testing, but until now has not been considered in the context of privacy. The problem of overfitting or "p-hacking" in the social and natural sciences has been referred to as the "statistical crisis in science" Gelman & Loken

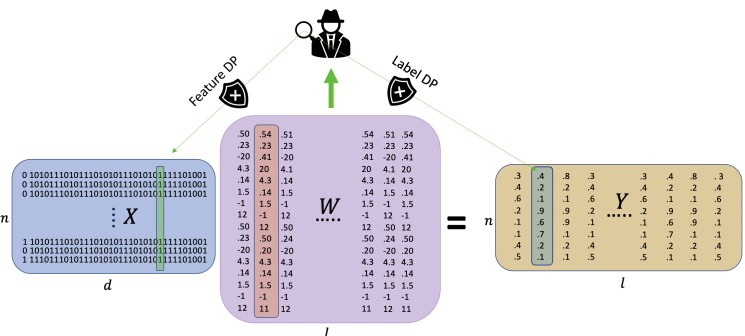

Figure 1: PRIMO imagines the scenario where the weights $W$ computed as a function of sensitive features $X$ and multiple outcomes $Y$ are published or leaked. Our algorithms prevent an adversary with access to $W$ from exposing the underlying sensitive data $x_i, y^i$.

(2014), and developing methods that quantify and mitigate the effects of overfitting has been the subject of much attention in the statistics and computer science communities Dwork et al. (2015a); Bassily et al. (2021); Korthauer et al. (2019). Given the ubiquity of Aspects 1 and 2, this raises an important question that we explore in this work:

**When computing $i = 1 \ldots l$ distinct regressions with a common set of $X$'s and distinct $y_i$'s, what is the optimal accuracy-privacy tradeoff?**

## 2 Preliminaries

We start by defining the standard linear regression problem. Let $X \in \mathcal{X}^n \subset \mathbb{R}^{n \times d}$ consist of $d$-dimensional samples from $n$ individuals, $y_i \in \mathcal{Y}^n$ be a vector of $n$ outcomes for a regression indexed by $i$, and parameter space $\mathcal{W} \subset \mathcal{R}^d$ denote the subset of linear regression coefficients we will optimize over.

For $w \in \mathcal{W}$ let $f(w) := \frac{1}{n} \sum_{k=1}^n (w \cdot x_k - y_{ik})^2$ be the linear regression objective, and denote by $w_{i*} = \arg\min_{w \in \mathbb{R}^d} f(w) = (X^T X)^\dagger X^T y_i$, where $\dagger$ is the Moore-Penrose inverse. For $\lambda > 0$ let $f_\lambda(w) = \frac{1}{n} \sum_{k=1}^n (w \cdot x_k - y_{ik})^2 + \lambda ||w||_2^2$ the ridge regression objective, and let $w_{i*}^\lambda = \arg\min f_\lambda(w)_{w \in \mathbb{R}^d} = (\frac{1}{n} X^T X + \lambda I_d)^{-1} \frac{1}{n} X^T y_i$.

The definition of data privacy we use throughout is the popular $(\varepsilon, \delta)$-differential privacy introduced in Dwork et al. (2006). We refer the reader the to Dwork & Roth (2014) for an overview of the basic properties of $(\varepsilon, \delta)-$DP including closure under post-processing and advanced composition. In order to define a differentially private mechanism, we first define what it means for two datasets to be adjacent.

**Definition 1.** *Two datasets $(X, y), (X', y') \in \mathcal{X}^n \times \mathcal{Y}^n$ are adjacent if they differ in a single element, e.g. there exists $j, j' \in [n]$, such that $X \cup \{x'_j\} \setminus \{x_j\} = X', y \cup \{y'_j\} \setminus \{y_j\} = y'$. We say that $(X, y), (X', y')$ are* feature-adjacent *if they share labels $y = y'$, and $\exists j \in [n]$ such that $X \cup \{x'_j\} \setminus \{x_j\} = X'$. Similarly, they are* label-adjacent *if $X = X'$, and $\exists j \in [n]$ such that $y \cup \{y'_j\} \setminus \{y_j\} = y'$.*

We denote adjacency of $(X, y), (X', y')$ by $(X, y) \sim (X', y')$.

**Definition 2.** *Let $\mathcal{M} : (\mathcal{X} \times \mathcal{Y})^n \to \mathcal{O}$ a randomized algorithm taking as input a dataset of $n$ records. $\mathcal{M}$ is $(\varepsilon, \delta)$-DP if $\forall$ pairs of adjacent datasets $(X, Y) \sim (X', Y'), O \subset \mathcal{O}$:*

$$\Pr[\mathcal{M}(X, Y) \in O] \leq e^\varepsilon \Pr[\mathcal{M}(X', Y') \in O] + \delta \tag{1}$$

When Definition 1 holds for adjacency defined over $(x_i, y_i)$ pairs, this is the standard setting of differential privacy, which we call "Full DP" in order to differentiate it from its relaxations. In the less restrictive case where Equation 1 holds only over pairs of label-adjacent datasets, we are in the well-studied setting of *label differential privacy* Ghazi et al. (2021); Esfandiari et al. (2021); Papernot et al. (2016). When Equation 1 holds only over pairs of feature-adjacent datasets, we say that we have *feature differential privacy*. We develop

specific PRIMO algorithms for these relaxations of DP in Section 5. We now formalize the *Private Regression In Multiple Outcomes (PRIMO)* problem.

**Definition 3.** *PRIMO. Let $x_i \in \mathcal{X} \subset \mathbb{R}^d, y_{ij} \in \mathcal{Y}$, for $i = 1 \ldots n, j = 1 \ldots l$. Let $X_{n \times d}$ the matrix with $i^{th}$ row $x_i$, and let $Y_{n \times l}$ the matrix with $j^{th}$ column $y_j = (y_{1j}, \ldots y_{nj})$. The optimal solution $W^*$ to the PRIMO problem is*

$$W^* = \underset{W \in \mathcal{W}^l \subset \mathbb{R}^{d \times l}}{\arg\min} ||XW - Y||_F^2$$

Given a randomized algorithm $\mathcal{M} : (\mathcal{X} \times \mathcal{Y}^l)^n \rightarrow \mathcal{W}^l$, we say that $\mathcal{M}$ is an $(\alpha, \beta, \varepsilon, \delta)$ solution to the PRIMO problem if (i) $\mathcal{M}$ is $(\varepsilon, \delta)$-DP, and (ii) with probability $1 - \beta$ over $\tilde{W} \sim \mathcal{M}$:

$$\frac{1}{nl}||X\tilde{W} - Y||_F^2 - \frac{1}{nl}||XW^* - Y||_F^2 < \alpha$$

We will use $y_j$ to denote the vector of $n$ outcomes for the $j^{th}$ outcome, and $y^i \in \mathcal{Y}^l$ for the vector of $l$ outcomes corresponding to individual $i \in [n]$. We note that in PRIMO under full DP, the adjacency condition holds over pairs $(x_i, y^i)$. The $(\epsilon, \delta)$-DP algorithm we will use most throughout is the Gaussian Mechanism. We state a version of the Gaussian mechanism with constant $c(\epsilon, \delta)$ that is valid for all $\epsilon > 0$, which follows from analyzing the mechanism using Renyi DP Mironov (2017) and converting back to $(\epsilon, \delta)$-DP.

**Lemma 1** (Dwork & Roth (2014)). *Let $f : \mathcal{X}^n \rightarrow \mathbb{R}^d$ an arbitrary d-dimensional function, and define it's sensitivity $\Delta_2(f) = \sup_{X \sim X'} ||f(X) - f(X')||_2$, where $X \sim X'$ are datasets that differ in exactly one element. Then the Gaussian mechanism `GaussMech`$(\varepsilon, \delta, \Delta)$ releases $f(X) + \mathcal{N}(0, \sigma^2)$, and is $(\varepsilon, \delta)$-differentially private for $\sigma \geq c(\varepsilon, \delta)\Delta_2(f)$, where $c(\varepsilon, \delta) = \sqrt{2(\frac{1}{\epsilon} + \frac{\log \frac{1}{\delta}}{\varepsilon^2})}$.*

Lastly, we define some notation. Given a vector $v$ and matrix $A$, $||v||_2, ||A||_2$ denote the $l_2$ and the spectral norm respectively, $||A||_F$ is the Frobenius norm, $|\mathcal{X}| = \sup_{x \in \mathcal{X}} ||x||_\infty, |\mathcal{Y}| = ||y||_\infty$, and $||\hat{w}||_2 = \sqrt{\frac{1}{l}||W^*||_F^2}$, where $W^*$ is the optimal PRIMO solution in Definition 3. Following convention in prior work, we will let $||\mathcal{X}||_2, ||\mathcal{Y}||_2, ||\mathcal{W}||_2$ denote $\sup_{x \in \mathcal{X}} ||x||_2, \sup_{y \in \mathcal{Y}} ||y||_2, \sup_{w \in \mathcal{W}} ||w||_2$ respectively.

## 2.1 Private Linear Regression

Private linear regression is well-studied under a variety of different assumptions on the data generating process and parameter regimes. Typically analysis of private linear regression is done either under the fully agnostic setting where only parameter bounds $||\mathcal{X}||_2, |\mathcal{Y}|, ||\mathcal{W}||_2$ are assumed, or under the assumption of a fixed design matrix and $y$ generated by a linear Gaussian model (the so-called realizable case), or under the assumption of a random design matrix from a known distribution Milionis et al. (2022). In this paper we focus on the first fully agnostic setting, because in our intended applications within the social and biomedical sciences in general we neither have realizability (when $y$ is actually a linear function of $x$) or Gaussian features. In the fully agnostic setting Wang (2018) provides a comprehensive survey of existing private regression approaches and bounds, including a discussion of the impact of different parameter norms.

Broadly speaking, techniques for private linear regression fall into 4 classes: sufficient statistics perturbation (`SSP`) Vu & Slavkovic (2009); Foulds et al. (2016), Objective Perturbation (`ObjPert`) Kifer et al. (2012), Posterior sampling Dimitrakakis et al. (2016), and privatized (stochastic) gradient descent (`NoisySGD`) Chaudhuri et al. (2011). The methods in this paper are a sub-class of `SSP`-based methods, which correspond to Algorithm 1 where $l = 1$. Methods based on `SSP` rely on perturbing $X^T X = \Sigma$ and $X^T y_i = \Sigma_{Xy}$ separately with matrices of Gaussian noise (denoted by $E_1, E_{2i}$ respectively), and then using these noisy estimates to compute the (regularized) least squares estimator:

$$\tilde{w}_{i*} = \underbrace{(X^T X + \lambda I + E_1)^{-1}}_{\text{noisy covariance term } \tilde{\Sigma}} \times \underbrace{(X^T y_i + E_{2i})}_{\text{noisy association term } \tilde{\Sigma}_{Xy}} \tag{2}$$

In their Theorem 5.3 Bassily et al. (2014) give a lower bound on the mean squared error of *any* $(\epsilon, \delta)$-DP algorithm for optimizing a Lipschitz objective function. In the case of linear regression (which corresponds to

PRIMO with $l = 1$), if $w_{private}$ denotes the output of any $(\epsilon, \delta)$-DP algorithm, then with probability at least $\frac{1}{3}$:

$$\alpha = f(w_{private}) - f(w_{i*}) \geq \min\{||\mathcal{Y}||_2^2, \frac{\sqrt{d}(||\mathcal{X}||_2^2||\mathcal{W}||_2^2 + ||\mathcal{X}||_2||\mathcal{W}||_2||\mathcal{Y}||_2}{n\epsilon}\} \tag{3}$$

Cai et al. (2020) prove a lower bound that is tailored for realizable low-dimensional $(n \geq d)$ linear regression, where $y \sim x^t\beta + N(0, \sigma^2)$, with $||x||_2 \leq 1$ and $||X^TX||_\infty = O(1/d)$, showing that $\alpha = \Omega(\sigma^2(\frac{d}{n} + \frac{d^2}{n^2\epsilon^2}))$. Neither of these bounds are exactly applicable to our setting; the bound of Bassily et al. (2014) is not neccessarily tight for linear regression, and the bound of Cai et al. (2020) makes realizability assumptions that clearly fail to hold for the type of genomic applications that motivate our work. We also note that all existing lower bounds for private linear regression are under full DP, and so do not immediately apply to the algorithms in Section 5 which operate in the weaker feature-DP or label-DP settings. In general, the focus of this work is on upper bounds, and so we highlight this curious lack of relevant lower bounds in the literature as a potential future direction, given the long line of work on upper bounds for private linear regression. Acknowledging the above issues with existing bounds, throughout the paper we'll use the lower bound of Bassily et al. (2014) as a benchmark for the cost to accuracy of taking $l > 1$, and in settings in which the bounds for PRIMO match this lower bound we will say we have "PRIMO for Free."

Given any $(\varepsilon, \delta)$-DP algorithm for computing $w_j$ privately, we can use it as a sub-routine to solve PRIMO by simply running it $l$ times to compute each column of $W$. Hence by running any of the optimal algorithms (for example SSP Vu & Slavkovic (2009)) $l$ times with parameters $\varepsilon' \approx \varepsilon/\sqrt{l}, \delta' \approx \delta/l$, by advanced composition for differential privacy Dwork & Roth (2014) we can achieve MSE, subject to $(\varepsilon, \delta)$-DP:

$$\alpha = \tilde{O}(\frac{\sqrt{ld}(||\mathcal{X}||_2^2||\mathcal{W}||_2^2 + ||\mathcal{X}||_2||\mathcal{W}||_2|\mathcal{Y}|}{n\varepsilon}), \tag{4}$$

where the $\tilde{O}$ hides factors of $\log(\frac{1}{\delta})$. So for a fixed privacy budget $\varepsilon$, this naive baseline is a factor of $\sqrt{l}$ worse than in the standard private regression setting where $l = 1$.

Further background on private query release for linear and low-sensitivity queries (relevant to our techniques in Section 5) under $l_\infty$ and $l_2$ error, and on sub-sampled linear regression (Subsection 7.7) is included in the Appendix.

## 3 Results

The primary contribution of this work is to introduce the novel Private Regression in Multiple Outcomes (PRIMO) problem, and to provide a class of algorithms that trade off accuracy, privacy, and computation. In addition to introducing the PRIMO problem, to our knowledge we are the first to apply private query release methods to linear regression (Section 5). We also provide a compelling practical application of our algorithms to the problem of genomic risk prediction with multiple phenotypes using data from The 1000 Genomes Project Fairley et al. (2019) and The Database of Genotypes and Phenotypes Mailman et al. (2007).

Algorithm 1 is our meta-algorithm for the PRIMO problem, and has two variants, `ReuseCovGauss` which corresponds to $\mathcal{M} = $ `GaussMech`, and `ReuseCovProj`, which corresponds to $\mathcal{M} = $ Algorithm 2. Algorithm 2 itself has two variants, which correspond to the label private setting (`GaussProj`$_X$), and to the feature private setting (`GaussProj`$_Y$) respectively. For any given setting of the problem parameters $(n, l, d, ||\mathcal{X}||_2, ||W||_2, |\mathcal{Y}|)$ one of these variants achieves superior performance, in terms of the upper bound we can prove on the error $\alpha$. Which variant achieves the lowest $\alpha$ depends on the value of $l$ relative to the other parameters, and the type of DP (Full, Feature, or Label) required. In Table 1 we summarize the theoretical upper bound on $\alpha$ achieved by the best PRIMO variant in each setting. In Sections 4, 5 we prove these upper bounds for each variant.

In Section 4 we adapt the previously proposed sufficient statistics perturbation (SSP) algorithm of Vu & Slavkovic (2009); Foulds et al. (2016) into the `ReuseCovGauss` Algorithm. Via a novel accuracy analysis of SSP for the case when the privacy levels for $E_1, E_{2i}$ differ in Equation 2 (Theorem 1), we show that since the noisy covariance matrix is reused across the regressions (as it only depends on $X$), by allocating the majority of our privacy budget to computing this term, we are able to obtain PRIMO for Free when $l$ is sufficiently

| Guarantees of ReuseCov | | | | |
|---|---|---|---|---|
| #$l$ of regressions | $\alpha =$ MSE | cost($l$) | $\mathcal{M}$ | Privacy |
| $l < \min(\frac{n}{\sqrt{d}}, \frac{\|\mathcal{X}\|_2^2\|\mathcal{W}\|_2^2}{|\mathcal{Y}|^2})$ | $\tilde{O}\left(\frac{\sqrt{d}\|\mathcal{X}\|_2^2\|\hat{w}\|_2^2}{n}\right)$ | 1 | Gauss | Full DP |
| $l \in (\frac{\|\mathcal{X}\|_2^2\|\mathcal{W}\|_2^2}{|\mathcal{Y}|^2}, \frac{n}{\sqrt{d}})$ | $\tilde{O}\left(\frac{\|\hat{w}\|_2\sqrt{ld}|\mathcal{Y}|\|\mathcal{X}\|_2}{n}\right)$ | $\sqrt{l}$ | Gauss | Full DP |
| $l > \frac{n}{\sqrt{d}}, n < \frac{\sqrt{d}\|\mathcal{X}\|_2^2\|\mathcal{W}\|_2^2}{|\mathcal{Y}|^2}$ | $\tilde{O}\left(\frac{\sqrt{d}\|\mathcal{X}\|_2^2\|\hat{w}\|_2^2}{n}\right)$ | 1 | Proj$_Y$ | Feature DP |
| $l > \frac{n}{\sqrt{d}}, n > \frac{\sqrt{d}\|\mathcal{X}\|_2^2\|\mathcal{W}\|_2^2}{|\mathcal{Y}|^2}$ | $\tilde{O}\left(\frac{\|\hat{w}\|_2 d^{1/4}|\mathcal{Y}|\|\mathcal{X}\|_2}{\sqrt{n}}\right)$ | $\frac{\sqrt{n}}{d^{1/4}}$ | Proj$_Y$ | Feature DP |
| $l > n^{\frac{2}{3}}$ | $\tilde{O}\left(\|\hat{w}\|_2 \frac{|\mathcal{X}|\|Y\|_2\sqrt{d}}{\sqrt{n}l^{1/4}}\right)$ | $l^{1/4}\sqrt{n}$ | Proj$_X$ | Label DP |

Table 1: In the table above cost($l$) is the ratio of the PRIMO MSE $\alpha$ to the lower bound in Equation 3, and $\mathcal{M}$ denotes the mechanism used to compute the association term $X^T Y$ in Algorithm 1.

small. This happens because the error term is dominated by the error in computing the noisy covariance matrix, which does not depend on $l$, rather than the error from computing the noisy association term (row 1 of Table 1). When $l > \frac{\|\mathcal{X}\|_2^2\|\mathcal{W}\|_2^2}{|\mathcal{Y}|^2}$, the asymptotic error of ReuseCovGauss is dominated by the error in computing the noisy association term, which has a $\sqrt{l}$ dependence in the error on $l$ (row 2 of Table 1). Given this, it is natural to ask if, under parameter regimes where the error from the association term dominates, can we obtain improved dependence of $\alpha$ on $l$ over the $\sqrt{l}$ given by ReuseCovGauss?

We answer this question in the affirmative in Section 5 with the introduction of the ReuseCovProj algorithms. The key idea is that instead of privately computing $\frac{1}{n}X^T Y$ via the Gaussian Mechanism (Lemma 1), which necessarily has error that scales like $\sqrt{l}$, in the setting where *either* $X$ or $Y$ are not considered private, projecting the noisy value $\frac{1}{n}X^T Y + \mathcal{N}(0, \sigma^2)$ returned by the Gaussian Mechanism onto the space of feasible values that the non-private quantity $\frac{1}{n}X^T Y$ could take (Line 5 of Algorithm 2), can reduce the error for sufficiently large $l$. This set of "feasible" values is the image of the domain of $Y$ under $X^T$ when $X$ is considered public, and it is the image of the domain of $X$ under $Y^T$ when $Y$ is considered public. The assumption that either $X$ or $Y$ are public are critical to these algorithms, as computing the projection depends on the space we are projecting into in a way that would not be easily privatized. The setting where $X$ is public and $Y$ is private corresponds to feature DP, and the setting where $X$ is public and $Y$ is public is label differential privacy. In rows $3-5$ in Table 1 we summarise the guarantees of Theorems 3,4 that characterize the accuracy in these settings. In both the Feature DP and Label DP settings, the projection improves the dependence in the error on $l, d$, at the cost of a factor of $\sqrt{n}$ in the denominator. As a result, until $l, d$ are sufficiently large relative to $n$ we'd expect the projection to actually increase error. We discuss these bounds further in Section 5.

In Section 6, we implement our ReuseCovGauss and ReuseCovProj algorithms using SNP data from two of the most common genomic databases Mailman et al. (2007); Fairley et al. (2019). We compare our algorithms to each other, and to the non-private baseline. In both settings we reach the surprising conclusion that even for relatively small values of $l = 11,100$, much smaller than the theory would predict, ReuseCovProj outperforms ReuseCovGauss. Moreover, ReuseCovProj is able to achieve non-trivial MSE (as measured by $R^2$) for very large values of $l$, as predicted by Theorem 3. We also compare both algorithms to the baseline of running a single private regression $l$ times (via DP-SGD Bassily et al. (2019)) and composing the privacy loss. DP-SGD performs significantly worse than any PRIMO algorithm unless we are in the setting of Row 2 in Table 1 where we expect no benefit from ReuseCovGauss or ReuseCovProj over SSP, consistent with the theory.

## 4 Full DP: The `ReuseCovGauss` Algorithm

We start by presenting our first algorithm for PRIMO, `ReuseCovGauss`, which corresponds to Algorithm 1 with $\mathcal{M} = $ `GaussMech`. To understand the intuition behind `ReuseCovGauss`, we start from the analysis of the ridge regression variant of `SSP` in Wang (2018). Equation 13 in Wang (2018) shows that if $\tilde{w}_i$ is the noisy ridge regressor output by `SSP`, and $w_{i*}$ is the (non-private) OLS estimator, then w.p. $1 - \rho$:

$$f(\tilde{w}_i) - f(w_{i*}) = \tilde{O}\left(\underbrace{\frac{d}{\lambda\epsilon^2}||\mathcal{X}||_2^2|\mathcal{Y}|^2}_{\text{association error term}} + \underbrace{\frac{d}{\lambda\epsilon^2}||\mathcal{X}||_2^4||\mathcal{W}||_2^2}_{\text{covariance error term}}\right) + \underbrace{\lambda||\mathcal{W}||_2^2}_{\text{error due to ridge penalty}} \tag{5}$$

Inspecting these terms, we see that when $||\mathcal{X}||_2||\mathcal{W}||_2 \gg ||\mathcal{Y}||_2$ the error is dominated by the covariance error term, which is the error due to random noise injected when privately computing $X^T X$, rather than the association error term, which is error due to random noise injected when privately computing $X^T y_i$. We now turn back to our PRIMO setting, and imagine independently applying `SSP` to solve each of our $i = 1 \ldots l$ regression problems. By the lower bounds in Bassily et al. (2014), given a fixed privacy budget this incurs at least a $\sqrt{l}$ multiplicative blow up in error. However, we notice that when our private linear regression subroutine is an `SSP` variant, this naive scheme of running $l$ independent copies of `SSP` is grossly wasteful. Since the $X$ matrix is shared across all the regressions, we can simply compute our noisy estimate of $X^T X$ once, and then reuse it across all the regressions. Combining these two observations, we present Algorithm 1. $\mathcal{M}$ can be any $(\epsilon/2, \delta/2)$-DP algorithm for estimating $X^T Y$, but when $M = $ `GaussMech` we call Algorithm 1 `ReuseCovGauss`.

---

**Algorithm 1** Input: $n, \lambda, X \in \mathcal{X}^n \subset \mathbb{R}^{d \times n}, Y = [y_1, \ldots y_l] \in \mathcal{Y}^{l \times n}$, privacy params: $\epsilon, \delta$

`ReuseCov`
1: Draw $E_1 \sim N_{d(d+1)/2}(0, \sigma_1^2)$, where $\sigma_1 = \frac{1}{n}c(\epsilon, \delta)||\mathcal{X}||_2^2$
2: Compute $\hat{I} = (\frac{1}{n}X^T X + E_1 + \lambda I)$
3: Draw $\hat{g} = [\hat{g}_1, \ldots \hat{g}_l] \sim \mathcal{M}(\epsilon/2, \delta/2, X, Y)$         ▷ $\mathcal{M} = $ `GaussMech` or Algorithm 2
4: **for** $i = 1 \ldots l$
5:      Set $\tilde{w}_i = \hat{I}^{-1}\hat{g}_i$         ▷ In practice computed via $\hat{I} = $ `QR` (Algorithm 3)
6: **end for**
7: Return $\tilde{W} = [\tilde{w}_1, \ldots \tilde{w}_l]$

---

**Theorem 1.** *With $\mathcal{M} = $ GaussMech$(\epsilon/2, \delta/2, \Delta = \frac{1}{n}\sqrt{l}||\mathcal{X}||_2|\mathcal{Y}|)$, Algorithm 1 is an $(\alpha, \rho, \epsilon, \delta)$ solution to the PRIMO problem with*

$$\alpha = \tilde{O}\left(||\hat{w}||_2\sqrt{\frac{d||\mathcal{X}||_2^4||\hat{w}||_2^2}{n^2} + \frac{ld|\mathcal{Y}|^2||\mathcal{X}||_2^2}{n^2}}\right), \tag{6}$$

*where $||\hat{w}||_2^2 = \frac{1}{l}||W^*||_F^2$, and $\tilde{O}$ omits terms polynomial in $\frac{1}{\epsilon}, \log(1/\delta), \log(1/\rho)$.*

*Proof.* The privacy proof follows from a straightforward application of the Gaussian mechanism. We note that releasing each $\hat{g}_i$ privately, is equivalent to computing $X^T Y + E_2$, where $E_2 \sim N_{d \times l}(0, \sigma_2^2)$. Now it is easy to compute $l$-sensitivity $\Delta(f)$ of $f(X) = X^T Y$. Fix an individual $i$, and an adjacent dataset $X' = X/\{x_i, y_i\} \cup \{x_i', y_i'\}$. Then $f(X) - f(X') = \Delta_V = [y_{i1}x_i - y_{i1}'x_{i1}', \ldots y_{il}x_i - y_{il}'x_{il}']$. Then:

$$||f(X) - f(X')||_2 = \sqrt{||\Delta_V||_F^2} = \sqrt{\sum_{j=1}^{l}||y_{ij}x_i - y_{ij}'x_{ij}'||_2^2} \leq \sqrt{l \cdot 4||\mathcal{X}||_2^2||\mathcal{Y}||_2^2} = 2\sqrt{l}||\mathcal{X}||_2||\mathcal{Y}||_2$$

Hence setting $\sigma_2 = c(\epsilon, \delta)2\sqrt{l}||\mathcal{X}||_2||\mathcal{Y}||_2/\epsilon$ by the Gaussian mechanism Dwork & Roth (2014) publishing $\hat{g}$ satisfies $(\epsilon/2, \delta/2) - DP$. Similarly if $g = X^T X$, $\Delta(g) \leq ||X||_2^2$, and so setting $\sigma_1 = c(\epsilon, \delta)||X||_2^2/\epsilon$, means publishing $\hat{I}$ is $(\epsilon/2, \delta/2)$-DP. By basic composition for DP, the entire mechanism is $(\epsilon, \delta)$-DP.

To prove the accuracy bound we follow the general proof technique developed in Wang (2018) analysing the accuracy guarantees of the ridge regression variant of SSP in the case $\lambda_{\min}(X^TX) = 0$, adding some mathematical detail to their exposition, and doing the appropriate book-keeping to handle our setting where the privacy level (as a function of the noise level) guaranteed by $E_1$ and $E_2$ differ. The reader less interested in these details can skip to Equation 11 below for the punchline.

Fix a specific index $i \in [l]$, and let $y = y_i$. We will analyze the prediction error of $\tilde{w}_i$ e.g. $f(\tilde{w}_i) - f(w_i^*)$. Then the following result is stated in Wang (2018) for which provide a short proof:

**Lemma 2** (Wang (2018))**.**

$$f(\tilde{w}_i) - f(w_{i*}) = \frac{1}{n}(||y - X\tilde{w}_i||^2 - ||y - Xw_{i*}||) = \frac{1}{n}||\tilde{w}_i - w_{i*}||^2_{X^TX}$$

*Proof.* We note that all derivatives of orders higher than 2 of $f(w) = \frac{1}{n}||y - Xw||^2$ are zero, and that $\nabla f_{w_{i*}} = 0$ by the optimality of $w_{i*}$. We also note that the Hessian $\nabla^2 f_w = \frac{1}{n}X^TX$ at all points $w$. Then by the Taylor expansion of $f(w)$ around $w_{i*}$:

$$f(\tilde{w}_i) = f(w_{i*}) + (\tilde{w}_i - w_{i*}) \cdot \nabla f_{w_{i*}} + \frac{1}{n}(\tilde{w}_i - w_{i*})'X^TX(\tilde{w}_i - w_{i*})$$

Which using $\nabla f_{w_{i*}} = 0$ and rearranging terms gives the result. $\square$

Now Corollary 7 in the Appendix of Wang (2018) states (without proof) the below identity, which we provide a proof of for completeness in Lemma 9 in the Appendix:

$$\tilde{w}_i - w_{i*} = (-X^TX + \lambda I + E_1)^{-1}E_1w_{i*} - \lambda(X^TX + \lambda I + E_1)^{-1}w_{i*} + (X^TX + \lambda I + E_1)^{-1}E_2 \quad (7)$$

Hence, still following Wang (2018), for any psd matrix $A$, $||\tilde{w}_i - w_{i*}||^2_A \leq$

$$3||(X^TX + \lambda I + E_1)^{-1}E_1w_{i*}||^2_A + 3\lambda^2||(X^TX + \lambda I + E_1)^{-1}w_{i*}||^2_A + 3||(X^TX + \lambda I + E_1)^{-1}E_2||^2_A \quad (8)$$

**Lemma 3** (Wang (2018))**.** *With probability* $1 - \rho$, $||E_1||_2 \leq (\lambda_{\min}(X^TX) + \lambda)/2$, *and hence*

$$X^TX + \lambda I + E_1 \succ .5(X^TX + \lambda I)$$

We also remark that $||By||^2_A = (By)^TABy = ||y||^2_{B^TAB}$ for any vector $y$, and matrices $A, B$.

Hence, Inequality 8, with $A = X^TX$ can be further simplified to:

$$3||(X^TX + \lambda I + E_1)^{-1}E_1w_{i*}||^2_A + 3\lambda^2||(X^TX + \lambda I + E_1)^{-1}w_{i*}||^2_A + 3||(X^TX + \lambda I + E_1)^{-1}E_2||^2_A \leq$$
$$O\left(||E_1w_{i*}||^2_{(X^TX + \lambda I + E_1)^{-1}} + \lambda^2||w_{i*}||^2_{(X^TX + \lambda I + E_1)^{-1}} + ||E_2||^2_{(X^TX + \lambda I + E_1)^{-1}}\right) \leq$$
$$\text{(Lemma 3)} \ O\left(||E_1w_{i*}||^2_{(X^TX + \lambda I)^{-1}} + \lambda^2||w_{i*}||^2_{(X^TX + \lambda I)^{-1}} + ||E_2||^2_{(X^TX + \lambda I)^{-1}}\right) \quad (9)$$

By basic properties of the trace we have: $\text{tr}((\lambda I + X^TX)^{-1}) \leq d\lambda_{max}(\lambda I + X^TX^{-1}) = \frac{d}{\lambda_{\min}(\lambda I + X^TX)} = \frac{d}{(\lambda_{\min}+\lambda)}$, and $||w_{i*}||^2_{(X^TX+\lambda I)^{-1}} \leq \frac{||w_{i*}||^2_2}{\lambda}$. Continuing from Wang (2018) by their Lemma 6, we can bound each $||E_1w_{i*}||^2_{(X^TX+\lambda I)^{-1}}$ and $||E_2||^2_{(X^TX+\lambda I)^{-1}}$.

**Lemma 4** (Wang (2018))**.** *Let* $w \in \mathbb{R}^d$ *and let* $E$ *a symmetric Gaussian matrix where the upper triangular region is sampled from* $N(0, \sigma^2)$ *and let* $A$ *be any psd matrix. Then with probability* $1 - \rho$:

$$||Ew||^2_A \leq \sigma^2 tr(A)||w||^2_2 \log(2d^2/\rho)$$

Then recalling that:

- $\sigma_1^2 = \tilde{O}(||\mathcal{X}||_2^4/\epsilon^2)$

- $\sigma_2^2 = \tilde{O}(l||\mathcal{X}||_2^2|\mathcal{Y}|^2/\epsilon^2)$

Plugging into Lemma 4, and bringing it all together we get:

$$n \cdot |f(\tilde{w}_i) - f(w_{i*})| \leq ||w_{i*} - \tilde{w}_i||_{X^TX} =$$
$$O\left(||E_1 w_{i*}||_{(X^TX+\lambda I)^{-1}}^2 + \lambda^2 ||w_{i*}||_{(X^TX+\lambda I)^{-1}}^2 + ||E_2||_{(X^TX+\lambda I)^{-1}}^2\right) =$$
$$\tilde{O}\left(\frac{d}{\lambda_{\min} + \lambda}||w_{i*}||_2^2(||\mathcal{X}||_2^4/\epsilon^2)\log(2d^2/\rho) + \lambda||w_{i*}||_2^2 + \frac{d}{\lambda_{\min} + \lambda}l||\mathcal{X}||_2^2|\mathcal{Y}|^2/\epsilon^2\log(2d^2/\rho)\right) \quad (10)$$

Upper bounding Equation 10 by taking $\lambda_{\min} = 0$, we minimize over $\lambda$ setting

$$\lambda = \tilde{O}\left(\frac{1}{\epsilon}\sqrt{d\log(2d^2/\rho)}||X||_2\sqrt{||\mathcal{X}||_2^2 + \frac{l|\mathcal{Y}|^2}{||w_{i*}||_2^2}}\right) \implies$$
$$|f(\tilde{w}_i) - f(w_{i*})| = \tilde{O}\left(\frac{1}{n\epsilon}\sqrt{d\log(2d^2/\rho)}||X||_2\sqrt{||\mathcal{X}||_2^2||w_{i*}||_2^4 + l|\mathcal{Y}|^2||w_{i*}||_2^2}\right) \quad (11)$$

Now if we are in the $\eta-$small regime, we have $|\mathcal{Y}| \leq \eta||\mathcal{X}||_2||w_{i*}||_2$, and so

$$l|\mathcal{Y}|^2 w_{i*}^2 \leq l\eta^2||\mathcal{X}||_2^2||w_{i*}||_2^4,$$

which reduces Equation 11 to:

$$|f(\tilde{w}_i) - f(w_{i*})| = \tilde{O}\left(\frac{1}{n\epsilon}\sqrt{d\log(2d^2/\rho)}||X||_2^2||w_{i*}||_2^2(\eta\sqrt{l})\right),$$

as desired.

$\square$

Inspecting Theorem 1, we see that when $l < \min(\frac{n}{\sqrt{d}}, \frac{||\mathcal{X}||_2^2||\mathcal{W}||_2^2}{|\mathcal{Y}|^2})$, Algorithm 1 with $\mathcal{M}$ the Gaussian mechanism achieves error $\tilde{O}\left(\frac{\sqrt{d}||\mathcal{X}||_2^2||\hat{w}||_2^2}{n}\right)$. Since this matches the lower bound in Equation 3 for a single private regression in this case we say that we have achieved "PRIMO for Free!"

## 5 Improved Algorithms for Large $l$

The improvement of `ReuseCovGauss` over the naive PRIMO baseline in the previous section only applies in the parameter regime where the asymptotic error in Equation 5 is dominated by the covariance error term. In the regime where the association error term dominates, `ReuseCovGauss` still incurs a $\sqrt{l}$ multiplicative factor in the error term, which does not improve over the baseline. In this section we show that if we are willing to relax from full DP to either feature DP or label DP, we can improve the dependence on $l$. Specifically, via a reduction from privately computing the association term to computing the image of a private vector under a public matrix, under feature DP we can obtain improved bounds for PRIMO when $l > \frac{n}{\sqrt{d}}$ (Subsection 5.1), and under *label differential privacy* we obtain improved bounds when $l > n^{2/3}$ (Subsection 5.2). In these cases, we obtain the surprising result that the MSE $\alpha$ has no explicit asymptotic dependence on $l$.

---

**Algorithm 2** Input: $X \in \mathcal{X}^n \subset (\mathbb{R}^d)^n$, $Y = [y_1, \ldots y_l] \in \mathcal{Y}^{l \times n}$, privacy params: $\epsilon, \delta$.

$\texttt{GaussProj}_Y$

1: Let $r = c(\epsilon, \delta) \frac{1}{n} \sup_i ||y^i||_2 ||\mathcal{X}||_2$
2: Sample $w \sim N(0,1)^{dl}$
3: Let $\tilde{g} = g + rw$, where $g = \frac{1}{n} X^T Y$            ▷ Up to this point this is just $\texttt{GaussMech}$
4: Formulate $C = C(Y) \in \frac{1}{n} \mathcal{Y}^{dl \times dn}$, $\text{vec}(X)$.         ▷ $C$ is only used in the projection step
5: Let $\hat{g} = \text{argmin}_{g \in K} ||g - \tilde{g}||_2^2$, where $K = C(\sqrt{n}||\mathcal{X}||_2 B_1)$.

---

## 5.1 PRIMO Under Feature Differential Privacy

Let us reconsider the problem of privately computing the association term $\frac{1}{n} X^T Y$ where $Y \in \mathcal{Y}^{n \times l}$, $X \in \mathcal{X}^n \subset \mathbb{R}^{n \times d}$, and $Y$ is considered public, so $\mathcal{M}$ is only constrained to be feature DP. Let $\text{vec}(X) = (x_{11}, \ldots, x_{1n}, x_{21}, \ldots x_{d1}, \ldots x_{dn}) \in \mathbb{R}^{nd}$. Then since $\frac{1}{n} X^T Y$ is *linear* in every entry of $X$, there exists a matrix $C(Y) \in \frac{1}{n} \mathcal{Y}^{dl \times dn}$ such that $\frac{1}{n} X^T Y = C \cdot \text{vec}(X)$. We explicitly derive an expression for $C$ in Subsection 7.4 of the Appendix. Note that $||\text{vec}(X)||_2^2 \leq n||\mathcal{X}||_2^2$, hence $\frac{1}{n} X^T Y \in C(\sqrt{n}||\mathcal{X}||_2 B_1)$, where $B_1 = \{x \in \mathbb{R}^{nd} : ||x||_2 \leq 1\}$.

Algorithm 2 is our projection-based subroutine for privately computing $\frac{1}{n} X^T Y$, which we call $\texttt{GaussProj}_Y$. We first state the squared error of $\texttt{GaussProj}_Y$ in Theorem 2, and then translate this error into the PRIMO error of $\texttt{ReuseCovProj}_Y$ (Algorithm 1 with $\mathcal{M}$ = Algorithm 2) in Theorem 3. The crux of the proof is Lemma 6 from Nikolov et al. (2013), which uses a geometric argument to bound the error after the projection.

**Theorem 2.** *Let $\hat{g}, g$ as in Line 4 of Algorithm 2. Then with probability $1 - \rho$ :*

$$||g - \hat{g}||_2^2 = O\left( \frac{l\sqrt{d}c(\epsilon, \delta)\sqrt{\log(2/\rho)}|\mathcal{Y}|^2||\mathcal{X}||_2^2}{n} \right)$$

*Proof.* $\mathcal{M}$ is $(\epsilon, \delta)$ differentially private by the Gaussian Mechanism and post-processing Dwork & Roth (2014). So we focus on the high probability accuracy bound. By Lemma 4 from Nikolov et al. (2013) we have, letting $K, r, w$ as defined in Algorithm 2, that:

$$||\hat{g} - g||_2^2 \leq 4||rw||_{K^\circ} = r \sup_{x \in K} x \cdot w \tag{12}$$

Using the fact that the $l_2$ norm is self-dual, we have:

$$||\hat{g} - g||_2^2 \leq 4r||w||_{K^\circ} \leq 4r||\mathcal{X}||_2 \sqrt{n} \sup_{z \in B_1} (Cz) \cdot w = 4r||\mathcal{X}||_2 \sqrt{n}||C^T w||_2 \tag{13}$$

So in order to bound $||\hat{g} - g||_2^2$ with high probability it suffices to bound $||C^T w||_2$ with high probability. This is the content of the Hanson-Wright Inequality for anisotropic random variables Vershynin (2019).

**Lemma 5** (Vershynin (2019)). *Let $C^T$ an $m \times n$ matrix, and $X \sim N(0,1)^n \in \mathbb{R}^n$. Then for a fixed constant $c > 0$:*

$$\mathbb{P}[\ |\ ||C^T X|| - ||C||_F\ | > t] \leq 2 exp(\frac{-ct^2}{||C||_F})$$

Lemma 5 shows that $||C^T w||_2 = O(||C||_F \sqrt{\log(2/\rho)})$ with probability $1 - \rho$. Given $x_{kj} \in X$, let $m = (k-1)n + j$ be the corresponding column of $C$, and let $c^m$ denote this column. Then $||c^m||_2 = \frac{1}{n}||(y_{1j}, \ldots y_{lj})||_2 = \frac{1}{n}||y^j||_2 \leq \frac{\sqrt{l}}{n}$. Hence $||C||_F = \frac{\sqrt{d \sum_{i=1}^n ||y^i||_2^2}}{n}$.

Plugging in the value of $r$ gives, with probability $1 - \rho$:

$$\frac{1}{dl}\mathbb{E}[||\hat{g} - g||_2^2] = O\left(\frac{1}{dl} \cdot \sqrt{n}||\mathcal{X}||_2 \frac{\sqrt{d\sum_{i=1}^{n}||y^i||_2^2}}{n}\sqrt{\log(2/\rho)} \cdot \frac{c(\epsilon, \delta)||\mathcal{X}||_2 \sup_i ||y^i||_2}{n}\right) \tag{14}$$

$$= O\left(\frac{c(\epsilon, \delta)\sqrt{\log(2/\rho)}\sqrt{\frac{1}{n}\sum_{i=1}^{n}||y^i||_2^2}\sup_i ||y^i||_2||\mathcal{X}||_2^2}{nl\sqrt{d}}\right), \tag{15}$$

Since both terms involving $Y$ in the numerator are $\leq \sqrt{l}|\mathcal{Y}|$ the bound follows. We note that since $Y$ is public, in practice we can compute these terms rather than using $|\mathcal{Y}|$, the worst case bound. $\qquad\square$

Notice that the mean squared error of the Gaussian mechanism without the projection is $O(r^2) = O(\frac{l||\mathcal{X}||_2^2|\mathcal{Y}|^2}{n^2})$, which for $l\sqrt{d} \gg n$ is strictly larger than the error of the projection mechanism. We also note that the bound in Theorem 2 is strictly better than the error given by applying the Median Mechanism algorithm of Blum et al. (2011) for low-sensitivity queries, which is tailored for $l_\infty$ error, and which also requires discrete $\mathcal{X}$. For example, when $\mathcal{Y} = \{0, 1\}, \mathcal{X} = \{0, 1\}^d$, then $|\mathcal{Y}| = 1, ||\mathcal{X}||_2 = \sqrt{d}$, and so Theorem 2 gives a bound of $\tilde{O}(\frac{\sqrt{d}}{n})$, whereas the Median Mechanism gives $\tilde{O}(\frac{d^{2/3}\log(dl)^2}{n^{2/3}})$ for the mean squared error. We now state the accuracy guarantees of $\texttt{ReuseCovProj}_Y$. The proof follows the proof of Theorem 1, substituting the bound from Theorem 2 into the $||E_2||_2^2$ term in Equation 7. We defer the full proof to the Appendix.

**Theorem 3.** *Let $\mathcal{A}$ denote the variant of Algorithm 1 with $\mathcal{M} = \texttt{GaussProj}_Y(X, Y, \epsilon/2, \delta/2)$. Then $\mathcal{A}$ is an $(\alpha, \rho, \epsilon, \delta)$ solution to the feature DP PRIMO problem with*

$$\alpha = \tilde{O}\left(||\hat{w}||_2\sqrt{\frac{d||\hat{w}||_2^2(||\mathcal{X}||_2^4)}{n^2} + \frac{\sqrt{d}|\mathcal{Y}|^2||\mathcal{X}||_2^2}{n}}\right),$$

*where $\tilde{O}$ omits terms polynomial in $\frac{1}{\epsilon}, \log(1/\delta), \log(1/\rho)$.*

Inspecting Theorem 3 in the $l > \frac{n}{\sqrt{d}}$ regime where the projection improves the error over the Gaussian Mechanism, when $n < \frac{\sqrt{d}||\mathcal{X}||_2^2||\mathcal{W}||_2^2}{|\mathcal{Y}|^2}$ is not too large, the dominant term in the error is $\tilde{O}\left(\frac{\sqrt{d}||\mathcal{X}||_2^2||\hat{w}||_2^2}{n}\right)$, which matches the lower bound 3, and so we achieve PRIMO for Free! When $n$ is sufficiently large the dominant error term is $\tilde{O}\left(\frac{||\hat{w}||_2 d^{1/4}|\mathcal{Y}|||\mathcal{X}||_2}{\sqrt{n}}\right)$ which is a factor of $\frac{n}{d^{1/4}}$ worse than the lower bound.

## 5.2 PRIMO Under Label Differential Privacy

The setting where $X$ is public and $Y$ is considered private is strictly easier to satisfy than setting in which they are both considered private, and in particular in order for Algorithm 1 to satisfy label DP, we only need to add noise to terms that involve $Y$. Thus we can compute $\hat{I}$ in Line 2 of Algorithm 1 exactly without adding any noise $E_1$. We still need to compute the association term $\frac{1}{n}X^TY$ privately in $Y$, where now the privacy guarantee holds over rows of $Y$ rather than rows of $X$. Computing $\frac{1}{n}X^TY$ privately is equivalent to computing $\frac{1}{n}Y_{l\times n}^T X_{n\times d}$, where we've added subscripts indicating the dimensions of the matrices. Then by direct analogy to the previous section, we can compute $\frac{1}{n}X^TY$ via Algorithm 2, where Theorem 2 holds by switching $X$ and $Y$, and $d, l$. We call this variant of Algorithm 2 $\texttt{GaussProj}_X$. Thus invoking Theorem 2 the error in computing the association term is with probability $1 - \rho$:

$$||g - \hat{g}||_2^2 = O\left(\frac{c(\epsilon, \delta)\sqrt{\log(2/\rho)}|\mathcal{X}|^2||\mathcal{Y}^l||_2^2 d\sqrt{l}}{n}\right) \tag{16}$$

The error in Equation 16 improves over the error given by $\mathcal{M} = \texttt{GaussMech}$ when $O(\frac{d\sqrt{l}}{n}) < O(\frac{dl^2}{n^2}) \implies l > n^{\frac{2}{3}}$, which we summarise in row 5 of Figure 1. Interestingly, in contrast to the public label setting, this condition has no dependence on the dimension $d$. Substituting Equation 16 into Equation 7 with $E_1 = 0$ since we are in the label-private setting and don't have to add noise to the covariance matrix, we can optimize over $\lambda$ to get:

**Theorem 4.** *Let $\mathcal{A}$ denote the label-private variant of Algorithm 1 with $E_1 = 0, \mathcal{M} = $ GaussProj$_X(X, Y, \epsilon/2, \delta/2)$. Then $\mathcal{A}$ is an $(\alpha, \rho, \epsilon, \delta)$ solution to the label DP PRIMO problem with*

$$\alpha = \tilde{O}\left(||\hat{w}||_2 \frac{|\mathcal{X}|||\mathcal{Y}^l||_2\sqrt{d}}{nl^{1/4}}\right)$$

We remark that while the form of Theorem 4 appears to suggest that the MSE decreases with increasing $l$, which is counterintuitive, $l$ actually appears in the numerator through $||\mathcal{Y}^l||_2 = O(|\mathcal{Y}|\sqrt{l})$, giving MSE with (mild) dependence $O(l^{1/4})$ on $l$.

### 5.3 Computational Efficiency

In this section we discuss the computational complexity of Algorithm 1 which can be broken down into 3 components:

- Step 1: Forming $X^T X$

- Step 2: In the case where $\mathcal{M}$ is the projection algorithm, Line 5 in Algorithm 2:

$$\hat{g} = [\hat{g}_1, \ldots \hat{g}_l] = \text{argmin}_{g \in C(\sqrt{n}||\mathcal{X}||_2 B_1)}||g - \tilde{g}||_2^2$$

- Step 3: the cost of computing $\hat{I}^{-1}\hat{g}_i = (X^T X + \lambda I + E_1)^{-1}\hat{g}_i \ \forall i \in [l]$

Step 1 forming the covariance matrix $X^T X$ is a matrix multiplication of two $d \times n$ matrices, which can be done via the naive matrix multiplication in time $O(nd^2)$, and via a long-line of "fast" matrix multiplication algorithms in time $O(d^{2+\alpha(n)})$; For example if $n < d^{.3}$ it can be done in time that is essentially $O(d^2)$ Gall (2012). Step 2 corresponds to minimizing a quadratic over a sphere. Setting $A = C^T C \in \mathbb{R}^{dn \times dn}, b = 2C^T\tilde{g} \in \mathbb{R}^{dl}$, then $\hat{g} = Cx$, where $x \in \mathbb{R}^{nd}$ is the minimizer of:

$$\min_{x \in B_1} x^t A x - b^t x \tag{17}$$

$$\text{s.t. } ||x||_2 \leq \sqrt{n}||\mathcal{X}||_2 \tag{18}$$

Now, given the spectral decomposition of $A = U\Lambda U^T$, and the coordinates of $b$ in the eigenbasis $U^T b$, Lemma 2.2 in Hager (2001) gives a simple closed form for $x$ that computes each coordinate in constant time. Since there are $nd$ coordinates of $x$, this incurs an additional additive factor of $O(nd)$ in the complexity, which is dominated by the cost of diagonalizing $A$. So the complexity of this step is the complexity of diagonalizing $A = C^T C$, or equivalently finding the right singular vectors of $C$, plus the complexity of computing $U^T b$. This is seemingly bad news, as $C \in \mathbb{R}^{dn \times dl}$ is a very high-dimensional matrix, and the complexity for computing the SVD of $C$ without any assumptions about its structure is $O(d^3 ln \min(l, n))$ Golub & Van Loan (1996). However, it is evident from the construction of $C$ in Subsection 5 that $C = I_d \otimes \frac{1}{n}Y^T$, where $\otimes$ is the Kronecker product. In Appendix 7.6 we show that using properties of the Kronecker product, we can compute the SVD$(C)$ in the same time as computing SVD$(Y)$, or $O(nl \min(n, l))$. $U^T b$ can be computed in time $O(nld)$ using similar tricks based on the Kronecker decomposition of $C$.

Finally, Step 3 can be completed by solving the equation $\hat{I}\hat{w}_i = \hat{g}_i, i = 1 \ldots l$ via the conjugate gradient method, which takes time $O(\gamma(\hat{I})d^2 \log(1/\epsilon))$ to compute an $\epsilon$-approximate solution Mahoney (2011) where $\gamma(\hat{I})$ is the condition number. We note that this has to be done separately for each $i = 1 \ldots l$ giving total time $O(l \cdot \gamma(\hat{I})d^2 \log(1/\epsilon))$. Alternatively, an exact solution $\hat{I}^{-1}\hat{g}_i$ can be computed directly using the QR decomposition of the matrix $\hat{I}$. The decomposition $\hat{I} = QR$ can be computed in time $O(d^3)$ Mahoney (2011) and does not depend on the $\hat{g}_i$, after which using $R\hat{w}_i = Q^T\hat{g}_i$, $\hat{w}_i$ can be computed in time $O(d^2)$ via backward substitution. This gives a total time complexity of $O(d^3 + ld^2)$. So if $d$ and $\frac{1}{\gamma(\hat{I})}$ are sufficiently small relative to $l$, e.g. if $l = \tilde{\Omega}(d/(\gamma(\hat{I})))$, it will be faster to use the $QR$ decomposition based method.

Putting this all together we have:

**Theorem 5.** *The complexity of Algorithm 3 is $O(\max(\min(nl^2, n^2l), nld, nd^2, ld^2, d^3))$.*

We summarize these algorithmic changes to `ReuseCovProj` in Algorithm 3. Theorem 5 shows that when $n > d > l$, the complexity of Algorithm 3 is $O(nd^2)$ or the cost of forming the covariance matrix. This motivates the results in Subsection 7.7 of the Appendix where we show how sub-sampling $s < n$ points and using matrix Chernoff bounds can improve this to $O(sd^2)$.

## 6 Experiments

**Datasets.** We evaluate the accuracy of our algorithms `ReuseCovGauss` and `ReuseCovProj`$_X$ on the task of genomic risk prediction using real $X$ data from two of the largest publicly available databases, and simulated outcomes $Y$ so that we can easily vary the number of outcomes $l$. In addition, in Appendix 7.8 we include experiments on two additional datasets over smaller values of $l$, one constructed by sub-sampling MNIST Deng (2012) and generating synthetic outcomes from a noisy linear model, and the other from entirely synthetic Gaussian data with outcomes generated either from a 2-layer MLP or a noisy linear model.

The genomic datasets are from two sources: the 1000 Genomes project (1KG) Fairley et al. (2019), and the Database of Genomes and Phenotypes Mailman et al. (2007) (accession phs000688.v1.p1). The datasets contain $n = 5008, 6042$ haplotypes at $d = 78961, 16721$ SNPs, respectively. Whenever experiments are run with a fixed value of $d < 78961, 16721$, we have randomly sub-sampled $d$ SNPs without replacement. In order to vary the number of phenotypes $l$ we generate synthetic phenotype data using our haplotype dataset. After centering our haplotype matrix $X$ by subtracting off the row means we generate synthetic phenotypes $y_i$ for $i = 1 \dots l$ by generating a random $\theta_i \sim \mathcal{N}(0, \frac{I_d}{\sqrt{d}}), y_{ij} \sim \theta_i \cdot x_j + \mathcal{N}(0, 1)$. In order to evaluate our algorithms at larger values of $n$ than exist in our genomic datasets, we train a GAN on the dbGaP dataset following Olagoke et al. (2023), and use it to generate synthetic samples to simulate having up to a million data points.

**Experimental Details.** For a given dataset and setting of $(n, d, l)$ we run `ReuseCovProj`$_X$ and `ReuseCovGauss` 10 different times, and calculate the resulting average MSE. We study two different regimes, one where $n$ is of moderate size corresponding to real genomic datasets, $d$ is small, and $l$ is varied from 1 to $1e5$ (Figure 2). The other regime models the scenario where in the near future genomic databases will have millions of individuals (e.g. UK Biobank already has $> 400000$ samples Privé (2022)), generating $n = 1e6$ samples from the GAN trained on dbGaP and again varying $l$ from 1 to $1e5$ (Figure 4). Note that we do not have experiments with larger values of $d$ for computational reasons; as discussed in Section 7.6 the runtime of both algorithms is quadratic in $d$.

Rather than plotting the MSE $\alpha$ directly, in Figure 2 we plot the coefficient of determination $R^2$, which is equivalent in that $R^2 = 1 - \frac{\alpha}{\frac{1}{nl} SS_{tot}}$. We use $R^2$ because it is easier to interpret given that achieving error better than the constant predictor implies $R^2 \in (0, 1]$. In Figures 3d,4 we also visualize the results using a different metric, the log of the ratio of the MSE of the private predictor to the MSE of the optimal non-private predictor, where a larger ratio indicates a greater multiplicative error.

**Realistic $n$, varying $l$.** We find that for values of $l$ starting as small as 11 for 1KG and 101 for dbGaP, `ReuseCovProj`$_X$ achieves higher $R^2$ than `ReuseCovGauss`, and is able to achieve non-trival $R^2$ for large $l$, which is consistent with our Theorem 3. For $l < 11$ on 1KG and $l < 101$ on dbGaP, `ReuseCovGauss` achieves higher $R^2$. In addition to being consistent with the theory, this is intuitive; when $l$ is small the bias induced by the projection outweighs the reduction in the overall noise. Figures 2a, 2b show that for $l > 101, 801$ on 1KG and dbGaP respectively `ReuseCovGauss` has negative $R^2$ – by contrast, zooming in on the $R^2$ curve for `ReuseCovProj` in Figures 2c, 2d we are able to achieve non-trivial $R^2$ at all values of $l$, and see minimal increase in error as $l$ increases.

**Large $n$, varying $l$.** Our upper bounds suggest that for large values of $n$, `ReuseCovGauss` will outperform `ReuseCovProj`$_X$, and that the MSE should increase polynomially with increasing $l$, or equivalently linearly in $\log l$ when we take the log ratio. Both of these trends can be seen in Figure 4(a). Given that the relative performance of `ReuseCovProj`$_X$ is improving as $l$ increases, we expect that for larger values of $l$ that were computationally prohibitive to run at such large $n$, `ReuseCovProj`$_X$ would again be the superior algorithm.

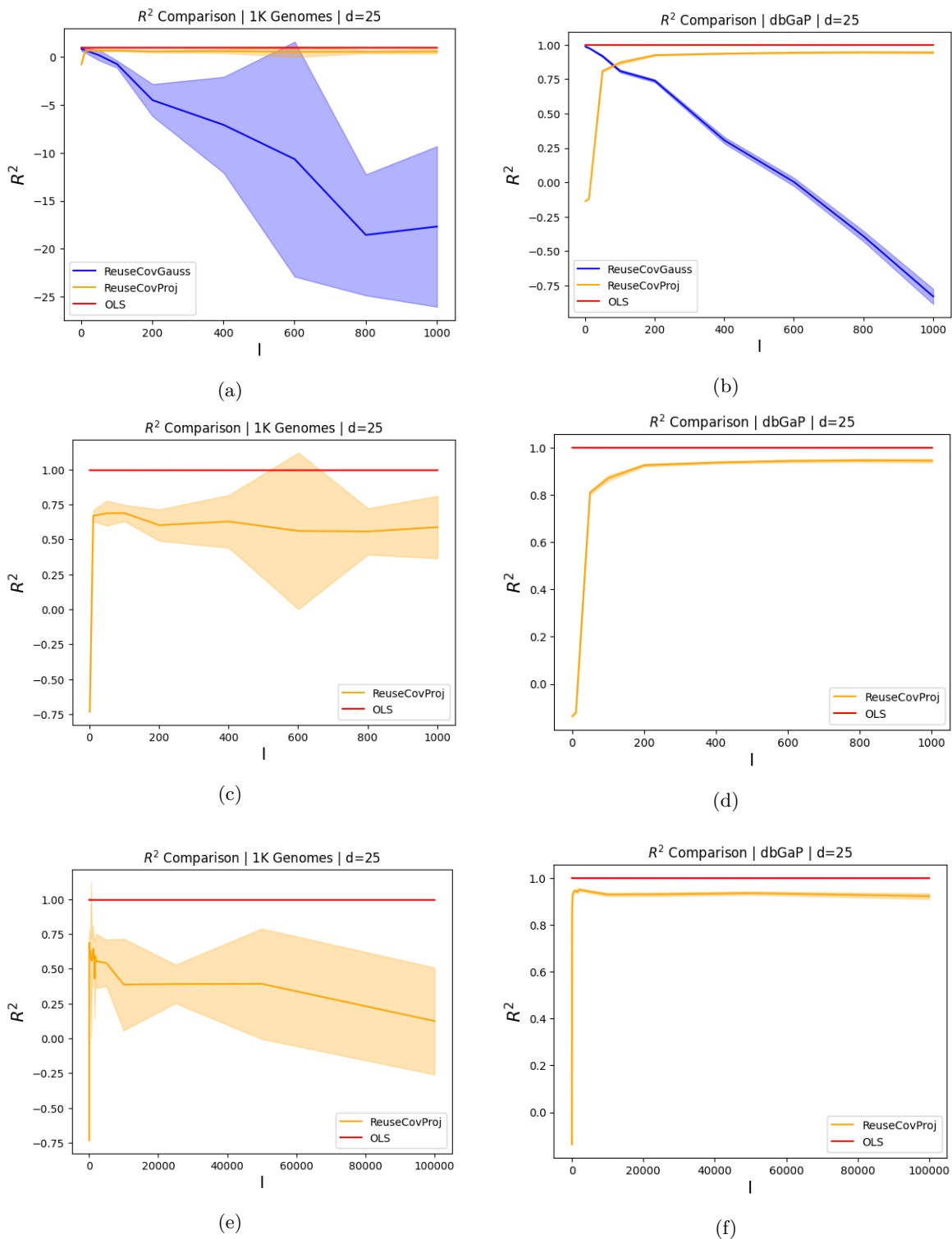

Figure 2: Each $R^2$ value is averaged over 10 iterations. The shaded area around the lines indicates the error bars for the $R^2$ value at a given value of $(l, d)$. In Figures (a)-(d) we plot the average $R^2$ for $d = 25, l = (1, 11, 101, 201, 401, 601, 801, 1001)$, fixing $(\epsilon, \delta) = (5, \frac{1}{n^2})$ with $n = 5008, 6042$. In Figures (e)-(f) we show $l$ up to $1e5$.

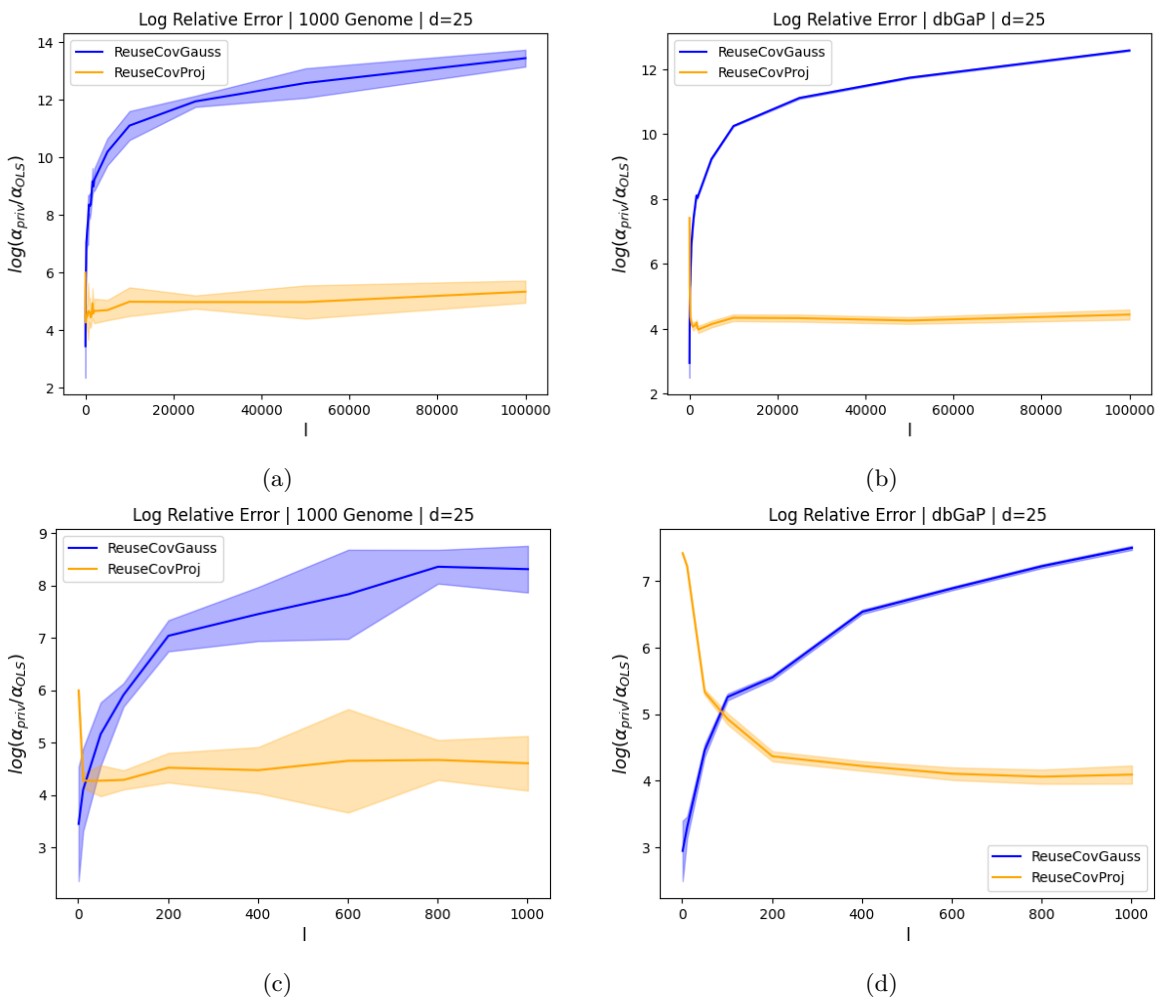

Figure 3: Comparing the log of the ratio of the squared loss of the private estimator to the square loss of the OLS estimator. Each value is averaged over 10 iterations. The shaded area around the lines indicates the error bars at the given value of $l$. (a) and (b) range $l$ up to $100,000$, (c) and (d) show $l = (1, 11, 101, 201, 401, 601, 801, 1001)$ while we fixed $d = 25$ and $(\epsilon, \delta) = (5, \frac{1}{n^2})$ with $n = 5008, 6042$ for 1000 Genomes and dbGaP respectively

**Additional Ablations.** In the setting where $l < \frac{n}{\sqrt{d}}$ (Row 2 of Table 1), theory suggests our algorithms should not outperform naive composition of private regressions. Since private regression methods like `DP-SGD` Bassily et al. (2019) are known to equal or outperform `SSP`, our `ReuseCov` algorithms which are variants of `SSP` might actually perform worse than composing `DP-SGD` across $l$ regressions. We construct such a dataset and show this is in fact the case in Figure 6 in the Appendix.

The experiments above on genomic data generate outcomes $Y$ from a noisy linear model; in Figure 5 in the Appendix we show that generating outcomes from a 2-layer MLP with noise doesn't change the relative ordering of the algorithms by performance.

## 6.1 Limitations

There are 3 main criticisms that can be levied at the results in this paper. The first is that the reductions in error in the feature DP and label DP settings due to projection in Lines $3-5$ of Figure 1 rely on sufficiently large $l : l > \frac{n}{\sqrt{d}}, l > n^{2/3}$ respectively. As a result, there are practical settings where $n$ is particularly large

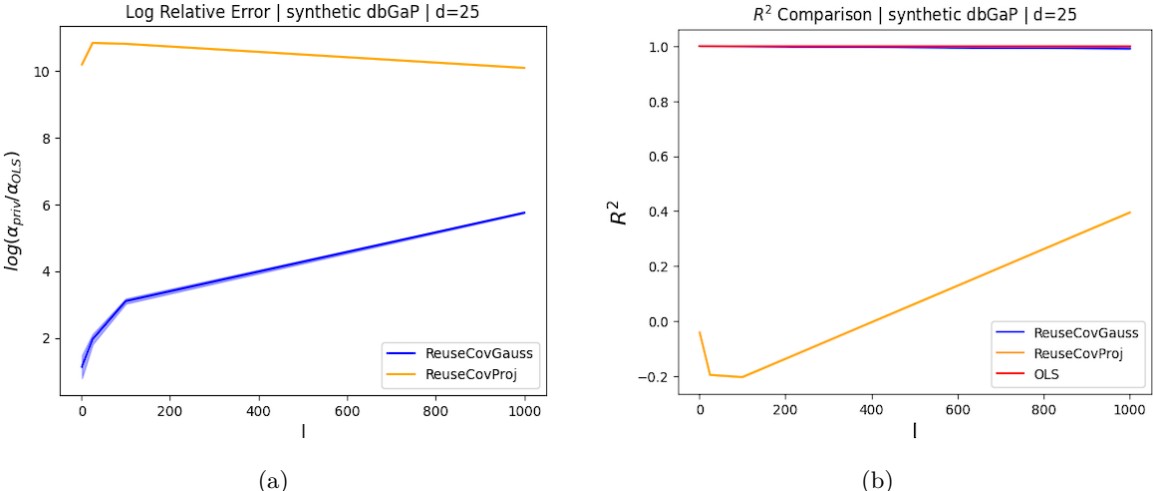

(a)                                                    (b)

Figure 4: (a) shows the log of the ratio of the squared loss of the private estimator to the square loss of the OLS estimator and (b) shows the $R^2$ values. For both of these, the value was averaged over 10 iterations. The shaded area around the lines indicates the error bars at a given value of $l$. We plot this for $l = (1, 10, 25, 100, 500, 1000, 5000)$ with $d = 25$, $(\epsilon, \delta) = (5, \frac{1}{n^2})$ and $n = 5 \cdot 1e5$ on a synthetic dbGaP dataset.

or $l$ is small where the theory does not suggest `ReuseCovProj` will outperform `ReuseCovGauss`. The second critique is that this improvement requires us relaxing our notion of DP to label or feature DP. In settings where both $X$ and $Y$ are highly sensitive, if we want to guarantee full DP `ReuseCovGauss` would be is our only option. Finally, all of our algorithms are variants of `SSP` Vu & Slavkovic (2009), which has complexity that scales super-linearly in $d$ due to forming the covariance matrix $X^T X$. For high-dimensional regression settings algorithms like private SGD are preferred due to their scalability Chaudhuri et al. (2011). With respect to the first critique, we note that in common settings, like in genomics, $d$ can be very large relative to $n$, and so $\frac{n}{\sqrt{d}}$ can be quite small in practice. For example, in the popular Fairley et al. (2019) dataset used in the experiments, $n = 5008$, and $d = 78961$, and so $\frac{n}{\sqrt{d}} \approx 17$. Similarly, if $n$ is of modest size, it is plausible that $l > n^{2/3}$. We also note that our experimental results show that the projection improves error at values of $l$ *much* smaller than the theory would suggest, making this algorithm the practical choice for values of $l$ as small as 11! Finally, the `ReuseCovGuass` Algorithm is itself also a contribution of this work. On the issue of scalability, in Subsection 5.3 we discuss how when $n > d > l$, the computational complexity of the `ReuseCovGauss` algorithm is the cost of computing the covariance matrix $X^T X$, which is $O(nd^2)$, which we improve to $O(sd^2)$ if we use $s$ points to sketch the covariance matrix (Appendix 7.7). Despite these improvements, the computational complexity of our methods is still super-linear in $d$.

Obtaining versions of PRIMO that scale for large values of $d$, and that do not require relaxing full DP in order to improve the dependence on $l$ as we do in Section 5, are exciting directions for future work.

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

# 7 Appendix

## 7.1 Additional Related Work

**Query Release.** In Subsection 5 we show that privately computing the association term $\frac{1}{n}X^TY$ is equivalent to the problem of differentially privately releasing a set of $l \cdot d$ low-sensitivity queries Bassily et al. (2021). While less is known about the optimal $l_2$ error for arbitrary low-sensitivity queries, it is clear that geometric techniques based on factorization and projection do not (at least obviously) apply, since there is no corresponding notion of the query matrix $A_Q$ Edmonds et al. (2019). In the case where $X, Y$ are both private, this corresponds to releasing a subset of $l \cdot d$ 2-way marginal queries over $d + l$ dimensions, which is well-studied Dwork et al. (2015b); Thaler et al. (2012); Ullman & Vadhan (2011). In the related work section of the Appendix we discuss why applying existing algorithms for private release of marginals seems unlikely be able to both improve over the Gaussian Mechanism and achieve non-trivial error. However, in the less restrictive but still practically relevant setting where the labels $Y$ are public, computing the association term $\frac{1}{n}X^TY$ is equivalent to the problem of differentially privately releasing a special class of $dl$ "low-sensitivity" queries we term "inner product queries" (Definition 6). For this special class of queries we can adapt the projected Gaussian Mechanism of Aydöre et al. (2021) that is optimal for linear queries under $l_2$ error in the sparse regime. We are able to obtain greatly improved results over the naive Gaussian Mechanism in the regime where $l > \frac{n}{\sqrt{d}}$, as summarised in Figure 1 and presented in Subsection 5. This is a key regime of interest, particularly for genomic data, where $d$ the number of SNPs could be in the hundreds of millions, $n$ the population size in the hundreds of thousands ($\sim 400K$ for UK Biobank Privé (2022) one of the most popular databases), and $l$ the number of recorded phenotypes could be in the thousands.

**Projection mechanisms and Algorithm 2.** Since we make heavy use of projection mechanisms in Subsection 5 in the *public label* setting, we elaborate on the difference between the existing methods and our Algorithm 2. The most technically related work to our Algorithm 2 is Aydöre et al. (2021), which is itself a variant of the projection mechanism of Nikolov et al. (2013). There are several key differences in our analysis and application.The projection in Nikolov et al. (2013) is designed for linear queries over a discrete domain, and runs in time polynomial in the domain size. Algorithm 2 allows continuous $\mathcal{X}, \mathcal{Y}$ and runs in time polynomial in $n, d, l$. Like the relaxed projection mechanism Aydöre et al. (2021), when our data is discrete we relax our data domain to be continuous in order to compute the projection more efficiently. Unlike in their setting which attempts to handle general linear queries, due to the linear structure of inner product queries our projection can be computed in polynomial time via linear regression over the $l_2$ ball, as opposed to solving a possibly non-convex optimization problem. Moreover, due to this special structure we can use the same geometric techniques as in Nikolov et al. (2013) to obtain theoretical accuracy guarantees (Theorem 2).

**Private release of 2-way marginals** Consider the problem of privately releasing all 2-way marginals over $n$ points in $\{0,1\}^{d+l}$. Theorem 5.7 in Dwork et al. (2015b) gives a polynomial time algorithm based on relaxed projections that achieves mean squared error $\tilde{O}(n\sqrt{d+l})$, which matches the best known information theoretic upper bound Dwork et al. (2015b), although there is a small gap to the existing lower bound $\min(n, (d+l)^2)$. This relaxed projection algorithm outperforms the Gaussian Mechanism when $n\sqrt{d+l} < ld \implies n < \frac{dl}{\sqrt{d+l}} \implies \frac{d\sqrt{d+l}}{n} > n$. Since the mean squared error of Algorithm 1 that used this projection as a subroutine is at least $\frac{d\sqrt{d+l}}{n}$, this means that in the regime where the projection outperforms the Gaussian Mechanism, we do not achieve mean squared error $< 1$ in our regression.

This suggests that, at least using the existing error analysis of SSP from Wang (2018), it seems unlikely that by applying specialized algorithms for private release of 2-way marginals to compute the $\frac{1}{n}X^TY$ term subject to differential privacy in $(X, Y)$, e.g. the label private setting, we can both improve over the Gaussian Mechanism and achieve non-trivial error.

**Linear Queries under $l_\infty$-loss.** Beyond 2-way marginals, the problem of privately releasing large numbers of linear queries (Definition 4) has been studied extensively. It is known that the worst case error is bounded by $\min(\frac{\sqrt{\log(|Q|)}(\log|\mathcal{X}|\log(1/\delta))^{1/4}}{\sqrt{n}\epsilon}, \frac{\sqrt{|Q|\log(1/\delta)}}{\epsilon n})$. The first term, which dominates in the so-called low-accuracy or "sparse" (Nikolov et al. (2013)) regime, is achieved by the *PrivateMultiplicativeWeights* algorithm of Hardt

& Rothblum (2010), which is optimal over worst case workloads Bun et al. (2013). However, this algorithm has running time exponential in the data dimension, which is unavoidable Vadhan (2017) over worst case $\mathcal{Q}$. The second term, which dominates for $n \gg \frac{|\mathcal{Q}|}{\log |\mathcal{Q}| \log |\mathcal{X}|^{1/4}}$, the "high accuracy" regime, is achieved by the simple and efficient Gaussian Mechanism Dwork & Roth (2014), which is also optimal over worst-case sets of queries $\mathcal{Q}$ Bun et al. (2013).

**Linear Queries under $l_2$-loss.** For the $l_2$ error, in the high accuracy $n \gg |\mathcal{Q}|$ regime the factorization mechanism achieves error that is exactly tight for any workload of linear queries $\mathcal{Q}$ up to a factor of $\log(1/\delta)$, although it is not efficient (Theorem Edmonds et al. (2019)). In the low-accuracy regime, the algorithm of Nikolov et al. (2013) that couples careful addition of correlated Gaussian noise (akin to the factorization mechanism) with an $l_1$-ball projection step achieves error within log factors of what is (a slight variant) of a quantity known as the *hereditary discrepancy* $\mathrm{opt}_{\epsilon,\delta}(A, n)$ (Theorem Nikolov et al. (2013)). This quantity is a known lower bound on the error of any $(\epsilon, \delta)$ mechanism for answering linear queries Muthukrishnan & Nikolov (2012), and so the upper bound is tight up to log factors in $|\mathcal{Q}|, |\mathcal{X}|$. Theorem 21 in Nikolov et al. (2013) analyzes the simple projection mechanism that adds independent Gaussian noise and projects rather than first performing the decomposition step that utilizes correlated Gaussian noise, achieving error $O(nd \log(1/\delta)\sqrt{\log |X|}/\epsilon)$, which matches the best known (worst case over $\mathcal{Q}$) upper bound for the sparse $n < d$ case Gupta et al. (2011). In our Theorem 2 we give such a universal upper bound, rather than one that depends on the hereditary discrepancy of the matrix $Y$. While the bound can of course be improved for a specific set of outcomes $Y$ by the addition of the decomposition step to the projection algorithm, we omit this step in favor of a simpler algorithm with more directly comparable bounds to existing private regression algorithms.

While the information-theoretic $l_\infty$ or $l_2$ error achievable for linear queries is well-understood Edmonds et al. (2019); Nikolov et al. (2013); Bun et al. (2013), as synthetic data algorithms like *PrivateMultiplicativeWeights* and *MedianMechanism*, or the factorization or projection mechanisms are in general inefficient, there are many open problems pertaining to developing efficient algorithms for specific query classes, or heuristic approaches that work better in practice. Examples of these approaches along the lines of the factorization mechanism McKenna et al. (2018; 2019), efficient approximations of the projection mechanism Aydöre et al. (2021); Dwork et al. (2015b), and using heuristic techniques from distribution learning in the framework of iterative synthetic data algorithms Yoon et al. (2019); Torkzadehmahani et al. (2020); Beaulieu-Jones et al. (2019); Neel et al. (2019).

**Sub-sampled Linear Regression.** In Subsection 7.7 we analyze SSP where we first sub-sample a random set of $s$ points without replacement, and use this sub-sample to compute the noisy covariance matrix. Sub-sampled linear regression has been studied extensively absent privacy, where it is known that uniform sub-sampling is sub-optimal in that it produces biased estimates of the OLS estimator, and performs poorly in the presence of high-leverage points Derezinski et al. (2018). To address these shortcomings, techniques based on leverage score sampling Drineas et al. (2006), volume-based sampling Derezinski & Warmuth (2017)Derezinski et al. (2018), and spectral sparsification Lee & Sun (2015) have been developed. Crucially, in these methods the probability of a point being sub-sampled is data-dependent, and so they are (not obviously) compatible with differential privacy.

## 7.2 Definitions

Throughout the paper we make heavy use of common matrix and vector norms. For a vector $v \in \mathbb{R}^d$ and matrix $A \in \mathbb{R}^{d \times d}$, $||v||_A^2 := v^t A v$, $||A||_2^2 = \lambda_{\max}(A^T A)$, $||v||_2^2 = \sum_{k=1}^d v_k^2$, $||A||_F^2 = \sum_{i,j \in [d]} a_{ij}^2$.

**Definition 4.** *Statistical Query Kearns (1993) Let $D \in \mathcal{X}^n$ a dataset. A linear query is a function $q : \mathcal{X} \to [0, 1]$, where $q(D) := \frac{1}{n} \sum_{x_i \in D} q(x_i)$.*

**Definition 5.** *Kasiviswanathan et al. (2010) Let $\mathcal{X} = \{0, 1\}^m$, and $\mathcal{Q}_k = \{q_{ij}\}_{1 \le i_1 < i_2 < i_k \le m}$, where $q_{ij}(x) := \prod_{u=1}^k x_{i_u}$. Then the class $\mathcal{Q}_k$ are called $k-$way marginals.*

Lastly, we define a new class of "low-sensitivity" queries Bassily et al. (2021) we call "inner product" queries that we will make heavy use of in Subsection 5.

**Definition 6.** *Inner Product Query. Let $\mathcal{X} \subset \mathbb{R}^d$ and $X \in \mathcal{X}^n \subset \mathbb{R}^{d \times n}$ a dataset with column $j$ corresponding to $x_j \in \mathcal{X}$. Let $y \in \mathbb{R}^n$, and $i \in [d]$. Then the pair $(i, y)$ defines an inner product query $q_{(i,y)} : \mathcal{X}^n \to \mathbb{R}$,*

$$q_{(i,y)}(X) := \frac{1}{n} \sum_{j=1}^{n} X_{ij} y_j = \frac{1}{n} e_i^T X y,$$

*where $e_i$ is the $i^{th}$ basis vector in $\mathbb{R}^d$.*

Changing one row $j$ of $X$, changes at most one summand $\frac{1}{n} X_{ij} y_j$ of $q_{(i,y)}(X)$, and hence the $l_2$ sensitivity $\Delta_2(q_{(i,y)}) \leq \frac{1}{n} |X_{ij}||y_j| \leq \frac{|\mathcal{X}||\mathcal{Y}|}{n}$, a fact we will use throughout.

### 7.3 Lemmas

**Lemma 6** (Raskutti et al. (2009)). *Let $K \subset \mathbb{R}^d$ be a symmetric convex body, let $g \in K$, and $\tilde{g} = g + w$ for some $w \in \mathbb{R}^d$. Then if $\hat{g} = argmin_{g' \in K} ||\tilde{g} - g'||_2^2$, then*

$$||\hat{g} - g||_2^2 \leq \min\{4||w||_2^2, 4||w||_{K^\circ}\}$$

**Lemma 7.** *Tropp (2010) Let $Z_i \in \mathbb{S}_d^+$, and sample $Z_1, \ldots Z_s$ without replacement from $\{Z_1, \ldots Z_n\}$. Suppose $\mathbb{E}[Z_i] = I_d$, and $\max_{i \in [n]} \lambda_{\max}(Z_i) \leq B$. Then for $\delta = \sqrt{\frac{2B \log(2d/\rho)}{s}}$, with probability $1 - \rho$:*

$$\lambda_{\max}\left(\frac{1}{s} \sum_{i=1}^{s} Z_i\right) < 1 + \delta, \quad \lambda_{\min}\left(\frac{1}{s} \sum_{i=1}^{s} Z_i\right) > 1 - \delta$$

**Lemma 8** (folklore e.g. Wang et al. (2018)). *Given a dataset $\mathcal{X}^n$ of $n$ points and an $(\epsilon, \delta)-DP$ mechanism $M$. Let the procedure **subsample** take a random subset of $s$ points from $\mathcal{X}^n$ without replacement. Then if $\gamma = s/n$, the procedure $M \circ$ **subsample** is $(O(\gamma\epsilon), \gamma\delta)-DP$ for sufficiently small $\epsilon$.*

The following lemma is used repeatedly in analyzing the accuracy of all `SSP` variants.

**Lemma 9.** *Let $A, B$ invertible matrices in $\mathbb{R}^{n \times n}$, and $v, c$ vectors $\in \mathbb{R}^n$.*

*Then*

$$A^{-1}v - (A + B)^{-1}(v + c) = (A + B)^{-1} B A^{-1} v - (A + B)^{-1} c$$

*Proof.* Expanding we get that:

$$A^{-1}v - (A + B)^{-1}(v + c) = (A^{-1} - (A + B)^{-1})v - (A + B)^{-1}c,$$

so it suffices to show that:

$$(A^{-1} - (A + B)^{-1})v = (A + B)^{-1} B A^{-1} v$$

Now the Woodbury formula tells us that $(A + B)^{-1} = A^{-1} - (A + AB^{-1}A)^{-1}$, hence

$$(A^{-1} - (A + B)^{-1})v = (A + AB^{-1}A)^{-1}v = (A(I + B^{-1}A))^{-1}v$$

Then since:

$$(A(I + B^{-1}A))^{-1} = (I + B^{-1}A)^{-1}A^{-1} = (B^{-1}(B + A))^{-1A^{-1}} = (B + A)^{-1} B A^{-1},$$

we are done. $\square$

### 7.4 Proofs from Subsection 5

**Computing the association term.** Consider entry $(k, j)$ of $\frac{1}{n} X^T Y$ for $k \in [d], j \in [l]$, $a_{kj} = (\frac{1}{n} X^T Y)_{kj} = e_k \frac{1}{n} X^T y_j = q_{(k,y_j)}(X)$, $q_{(k,y_j)}$ is the inner product query of Definition 6. Thus privately computing $\frac{1}{n} X^T Y$, is equivalent to privately releasing answers to $dl$ inner product queries $\{q_{(k,y_j)}(X)\}_{k \in [d], j \in [l]}$.

It will be convenient for us to write $q_{(k,y_j)}(X)$ as a single inner product. Let $\text{vec}(X) = (x_{11}, \ldots, x_{1n}, x_{21}, \ldots x_{d1}, \ldots x_{dn}) \in \mathbb{R}^{nd}$, and given $y = \text{vec}(X)$, let $\text{mat}(y) = X$. Denote by $c_{kj} \in \mathbb{R}^{nd}$ the vector that has all zeros except in positions $(k-1)n+1, \ldots kn$ it contains $\frac{1}{n} y_{1j}, \ldots \frac{1}{n} y_{nj}$. Then it is clear that $q_{(k,y_j)}(X) = c_{kj}^T \text{vec}(X)$, so if we let $C \in (\frac{1}{n} \mathcal{Y})^{dl \times dn}$ be the matrix with row $kj \in [dl]$ equal to $c_{kj}$, then $\frac{1}{n} X^T Y = C \cdot \text{vec}(X)$.

**Proof of Theorem 4:**

*Proof.* We start with our usual expansion of $f(\tilde{w}_i) - f(w_{i*})$, up until Equation 10, we have with probability $1 - \rho$ for every $i \in [l]$:

$$n \cdot f(\tilde{w}_i) - f(w_{i*}) = \tilde{O}\left(\frac{d}{\lambda_{\min} + \lambda} ||w_{i*}||_2^2 (||\mathcal{X}||_2^4/\epsilon^2) \log(2d^2/\rho) + \lambda ||w_{i*}||_2^2 + \frac{1}{\lambda_{\min} + \lambda} ||E_{2i}||_2^2\right) \quad (19)$$

Aggregating over $i$ and rearranging gives:

$$\frac{n}{l} \sum_{i=1}^{l} f(\tilde{w}_i) - f(w_{i*}) = \tilde{O}\left(\frac{d}{\lambda_{\min} + \lambda} ||\tilde{w}||_2^2 (||\mathcal{X}||_2^4/\epsilon^2) \log(2d^2/\rho) + \lambda ||\tilde{w}||_2^2 + \frac{1}{\lambda_{\min} + \lambda} \frac{1}{l} \sum_{i=1}^{l} ||E_{2i}||_2^2\right), \quad (20)$$

where $||\tilde{w}||_2^2 = \frac{1}{l} \sum_{i=1}^{l} ||w_{i*}||_2^2 = \frac{1}{l} ||W^*||_F^2$. Then by Theorem 2, we have with $\frac{1}{l} \sum_{i=1}^{l} ||E_{2i}||_2^2 = O\left(c(\epsilon, \delta) \sqrt{\log(2/\rho)} n \sqrt{d} |\mathcal{Y}|^2 ||\mathcal{X}||_2^2\right)$ with probability $1 - \rho$. So with probability $1 - 2\rho$, we have $\frac{n}{l} \sum_{i=1}^{l} f(\tilde{w}_i) - f(w_{i*}) =$

$$\tilde{O}\left(\frac{d}{\lambda_{\min} + \lambda} ||\tilde{w}||_2^2 (||\mathcal{X}||_2^4/\epsilon^2) \log(2d^2/\rho) + \lambda ||\tilde{w}||_2^2 + \frac{1}{\lambda_{\min} + \lambda} c(\epsilon, \delta) \sqrt{\log(2/\rho)} n \sqrt{d} |\mathcal{Y}|^2 ||\mathcal{X}||_2^2\right) \quad (21)$$

Finally optimizing over $\lambda$ gives the desired result:

$$\alpha = \tilde{O}\left(||\tilde{w}||_2 \sqrt{\frac{d ||\tilde{w}||_2^2 (||\mathcal{X}||_2^4/\epsilon^2) \log(2d^2/\rho)}{n^2} + \frac{c(\epsilon, \delta) \sqrt{\log(2/\rho)} \sqrt{d} |\mathcal{Y}|^2 ||\mathcal{X}||_2^2}{n}}\right)$$

$\square$

### 7.5 Proofs from Subsection 7.7

**Proof of Theorem 6**:

*Proof.* Our analysis will hinge on the case where $l = 1$ e.g. that of standard private linear regression, which we will extend to the PRIMO case by our choice of $\epsilon$ as in the proof of Theorem 1. The fact that the Algorithm is $(O(\epsilon), \delta)$ private follows immediately from the Gaussian mechanism, and the secrecy of the sub-sample lemma (Lemma 8), which is why we can set $\epsilon_1 = \frac{n}{s} \epsilon/2$ in Line 2. We proceed with the accuracy analysis.

Define:

- $w_{i*} = (X^T X)^{-1} X^T Y$: the least squares estimator as before
- $w_{i*}^\lambda = (\frac{1}{n} X^T X + \lambda I)^{-1} \frac{1}{n} X^T Y$: the ridge regression estimator

- $w_s = (\frac{1}{s}X_S^T X_S + \lambda I)^{-1}(\frac{1}{n}X^T Y)$: the sub-sampled least squares estimator

- $\tilde{w}_s = (\frac{1}{s}X_S^T X_S + E_1 + \lambda I)^{-1}(\frac{1}{n}X^T Y + E_2)$: differentially private estimate of $w_s$

We note that $0 \le f(w_{i*}^\lambda) - f(w_{i*}) \le \lambda \left(||w_{i*}||_2^2 - ||w_{i*}^\lambda||_2^2\right) \le \lambda||\mathcal{W}||_2^2$. Then by the Lemma 2 and Cauchy-Schwartz with respect to the norm $||\cdot||_{\frac{X^T X}{n}}$:

$$|f(\tilde{w}_s) - f(w_{i*})| = ||\tilde{w}_s - w_{i*}||_{\frac{X^T X}{n}}^2 \le$$
$$3||w_{i*} - w_{i*}^\lambda||_{\frac{X^T X}{n}}^2 + 3||w_{i*}^\lambda - w_s||_{\frac{X^T X}{n}}^2 + 3||w_s - \tilde{w}_s||_{\frac{X^T X}{n}}^2 \le$$
$$3\lambda||\mathcal{W}||_2^2 + 3||w_s - w_{i*}^\lambda||_{\frac{X^T X}{n}}^2 + 3||\tilde{w}_s - w_s||_{\frac{X^T X}{n}}^2 \quad (22)$$

Lemmas 14, 13 bound these terms with high probability.

**Lemma 10.** *Under the assumption $||\mathcal{X}||_2 = O(n\lambda)$, then w.p. $1 - \rho/2$:*

$$||w_{i*}^\lambda - w_s||_{\frac{X^T X}{n}} = O\left(\frac{||\mathcal{X}||_2|\mathcal{Y}|\log(2d/\rho)}{\lambda s}\right)$$

**Lemma 11.** *Under the assumption $||\mathcal{X}||_2 = O(n\lambda)$, then w.p. $1 - \rho/2$:*

$$||w_s - \tilde{w}_s||_{\frac{X^T X}{n}} = O(\frac{||X||_2^4}{n^2\epsilon^2}||\mathcal{W}||_2^2 \cdot \frac{d}{\lambda}\frac{\log(2d/\rho)}{s} + \frac{l||\mathcal{X}||_2^2||\mathcal{Y}||_2^2}{n^2\epsilon^2} \cdot \frac{d}{\lambda}\frac{\log(2d/\rho)}{s}) \quad (23)$$

Then by Equation 35 and Lemmas 14, 13, we get that w.p. $1 - \rho$:

$$f(\tilde{w}_s) - f(w_{i*}) =$$
$$O\left(\lambda||w_{i*}||_2^2 + \frac{||\mathcal{X}||_2|\mathcal{Y}|}{\lambda}(\frac{\log(2d/\rho)}{s}) + \frac{||X||_2^4 n^2}{s^4\epsilon^2}||\mathcal{W}||_2^2 \cdot \frac{d}{\lambda}(1 + \frac{\log(2d/\rho)}{s}) + \frac{l||\mathcal{X}||_2^2||\mathcal{Y}||_2^2}{n^2\epsilon^2} \cdot \frac{d}{\lambda}(1 + \frac{\log(2d/\rho)}{s})\right),$$
$$(24)$$

Summing over $i$ and minimizing over $\lambda$ we set

$$\lambda = \frac{\sqrt{||\mathcal{X}||_2|\mathcal{Y}|(\frac{\log(2d/\rho)}{s}) + \frac{||X||_2^4 n^2}{s^4\epsilon^2}||\mathcal{W}||_2^2 \cdot \frac{d}{\lambda}(1 + \frac{\log(2d/\rho)}{s}) + \frac{l||\mathcal{X}||_2^2||\mathcal{Y}||_2^2}{n^2\epsilon^2} \cdot \frac{d}{\lambda}(1 + \frac{\log(2d/\rho)}{s})}}{\sqrt{1}l||W||_F},$$

which completes the result. $\qquad\square$

**Proof of Lemma 14**:

*Proof.* Now:
$$||w_{i*}^\lambda - w_s||_{\frac{1}{n}X^T X}^2 = n||w_{i*}^\lambda - w_s||_{\frac{1}{n}X^T X}^2 \le n||w_{i*}^\lambda - w_s||_{\frac{1}{n}X^T X + \lambda I}^2$$

We will focus on $||w_{i*}^\lambda - w_s||_{\frac{1}{n}X^T X + \lambda I}^2$. Let $\Sigma = \frac{1}{n}X^T X + \lambda I$, $\Sigma_s = \frac{1}{s}\sum_{j\in S}x_s x_s^T + \lambda I$, and $v = \frac{1}{n}X^T Y$. Expanding $||w_{i*}^\lambda - w_s||_\Sigma =$

$$(\Sigma_s^{-1}v - \Sigma^{-1}v)^T \Sigma(\Sigma_s^{-1}v - \Sigma^{-1}v) = v^T \overbrace{(\Sigma_s^{-1} - \Sigma^{-1})\Sigma(\Sigma_s^{-1} - \Sigma^{-1})}^{A} v = v^T A v \quad (25)$$

Now since $A$ is Hermitian, we know $||v||_A \le ||v|||A||_2$. Since $||v|| \le ||\mathcal{X}||_2|\mathcal{Y}|$, it suffices to bound $||A||_2$ with high probability. Noting that $A = (\Sigma_s^{-1} - \Sigma^{-1})\Sigma(\Sigma_s^{-1} - \Sigma^{-1}) = \Sigma_s^{-1}\Sigma^{1/2}(I - \Sigma^{-1/2}\Sigma_s\Sigma^{-1/2})(I - \Sigma^{-1/2}\Sigma_s\Sigma^{-1/2})\Sigma^{1/2}\Sigma_s^{-1}$, we have by the sub-multiplicativity of the operator norm:

$$||A||_2 \leq \left( ||\Sigma_S^{-1}\Sigma^{1/2}||_2^2 \right) \cdot \left( ||I - \Sigma^{-1/2}\Sigma_s\Sigma^{-1/2}||_2^2 \right) \leq$$

$$\frac{\lambda_{\max}(\Sigma)}{\lambda^2} \cdot \left( ||I - \Sigma^{-1/2}\Sigma_s\Sigma^{-1/2}||_2^2 \right) \quad (26)$$

Now consider $\Sigma^{-1/2}\Sigma_s\Sigma^{-1/2} = \Sigma^{-1/2}\frac{1}{s}\sum_{j\in S}(x_sx_s^T + \lambda I)\Sigma^{-1/2} = \frac{1}{s}\sum_{j\in s}Z_i$. Then note that $\mathbb{E}[Z_i] = \Sigma^{-1/2}(\frac{1}{n}\mathbb{E}[x_ix_i^T] + \lambda I)\Sigma^{-1/2} = \Sigma^{-1/2}\Sigma\Sigma^{-1/2} = I$, and that $\lambda_{max}(Z_i) \leq ||\Sigma^{-1}||_2||\frac{1}{n}x_ix_i^T + \lambda I||_2 \leq 1 + \frac{||\mathcal{X}||_2}{n\lambda}$. Now we can bound $||I - \frac{1}{s}\sum_{j\in s}Z_i||_2$ by Theorem 2.2 in Tropp (2010):

**Lemma 12.** *Tropp (2010) Let $Z_1, \ldots Z_s$ sampled without replacement from $\{Z_1, \ldots Z_n\}$. Then if $Z_i \in \mathbb{S}_d^+, \mathbb{E}[Z_i] = I_d$, and $\max_{i\in[n]}\lambda_{\max}(Z_i) \leq B$ w.p. $1 - \rho$, for $\delta = \sqrt{\frac{2B\log(2d/\rho)}{s}}$:*

$$\lambda_{\max}(\frac{1}{s}\sum_{i=1}^s Z_i) < 1 + \delta, \quad \lambda_{\min}(\frac{1}{s}\sum_{i=1}^s Z_i) > 1 - \delta$$

So by Lemma 12, we know that with probability $1 - \rho$: $|\lambda_{\min}(I - \Sigma^{-1/2}\Sigma_s\Sigma^{-1/2})| = 1 - \lambda_{\max}(\Sigma^{-1/2}\Sigma_s\Sigma^{-1/2}) \leq \delta$, and similarly $|\lambda_{\max}(I - \Sigma^{-1/2}\Sigma_s\Sigma^{-1/2})| \leq \delta$, thus $||I - \Sigma^{-1/2}\Sigma_s\Sigma^{-1/2})||_2 \leq \delta$. Substituting this all into Equation 26 and noting $\lambda_{\max}(\Sigma) \leq \frac{||\mathcal{X}||_2}{n} + \lambda$ we get with probability $1 - \rho$:

$$||w_{i*}^\lambda - w_s||_{\frac{1}{n}X^TX} \leq ||\tilde{w}_s - w_s||_\Sigma \leq ||\mathcal{X}||_2|\mathcal{Y}|n||A||_2 \leq ||\mathcal{X}||_2|\mathcal{Y}|\frac{\lambda_{\max}(\Sigma)}{\lambda^2}\delta^2 = \quad (27)$$

$$2||\mathcal{X}||_2|\mathcal{Y}|\frac{(\frac{||\mathcal{X}||_2}{n} + \lambda)^2}{\lambda^3}\frac{\log(2d/\rho)}{s}, \quad (28)$$

which under the assumption $||\mathcal{X}||_2 = O(n\lambda)$ gives $O(\frac{||\mathcal{X}||_2|\mathcal{Y}|\log(2d/\rho)}{\lambda s})$. as desired. $\square$

**Proof of Lemma 13:**

*Proof.* By Lemma 9, with $A = \frac{1}{s}X_S^TX + \lambda I, B = E_1, c = E_2, v = \frac{1}{n}^TY$, we get $w_s - \tilde{w}_s = (\frac{1}{s}X_S^TX + \lambda I + E_1)E_1w_s - (\frac{1}{s}X_S^TX + \lambda I + E_1)^{-1}E_2$, and so

$$||w_s - \tilde{w}_s||_{\frac{X^TX}{n}}^2 \leq 2||(\frac{1}{s}X_S^TX + \lambda I + E_1)^{-1}E_1w_s||_{\frac{X^TX}{n}}^2 + 2||(\frac{1}{s}X_S^TX + \lambda I + E_1)^{-1}E_2||_{\frac{X^TX}{n}}^2$$

Under the assumption $||E_1||_2 \leq \lambda/2$, this becomes

$$||w_s - \tilde{w}_s||_{\frac{X^TX}{n}}^2 = O\left( ||E_1w_s||_{(\frac{1}{s}X_S^TX_S + \lambda I)^{-1}(X^TX/n + \lambda I)(\frac{1}{s}X_S^TX_S + \lambda I)^{-1}} \right) +$$

$$O\left( ||E_2||_{(\frac{1}{s}X_S^TX_S + \lambda I)^{-1}(X^TX/n + \lambda I)(\frac{1}{s}X_S^TX_S + \lambda I)^{-1}} \right) \quad (29)$$

Now to apply Lemma 4, we need to bound

$$\text{Tr}((\frac{1}{s}X_S^TX_S + \lambda I)^{-1}(X^TX/n + \lambda I)(\frac{1}{s}X_S^TX_S + \lambda I)^{-1}) \leq d\lambda_{\max}(\Sigma_S)^{-1}(\Sigma)(\Sigma_S)^{-1}) =$$

$$d\lambda_{\max}(\Sigma^{-1/2}(\Sigma^{1/2}\Sigma_S^{-1}\Sigma^{1/2})^2\Sigma^{-1/2}) \leq$$

$$d\lambda_{\max}(\Sigma^{-1})\frac{1}{\lambda_{\min}(\Sigma^{-1/2}\Sigma^{1/2}\Sigma_S^{-1}\Sigma^{1/2})}^2 \leq \frac{d}{\lambda}(\frac{1}{1-\delta})^2, \quad (30)$$

where the last inequality follows from Lemma 12. Applying Lemma 4 we get that with probability $1 - 2\rho$:

$$||w_{i*}^\lambda - w_s||_{X^TX/n} = O\left( \sigma_1^2 \cdot \frac{d}{\lambda}(\frac{1}{1-\delta})^2 \cdot ||w_s||_2^2\log(2d^2/\rho) + \sigma_2^2 \cdot \frac{d}{\lambda}(\frac{1}{1-\delta})^2\log(2d^2/\rho) \right), \quad (31)$$

From Lemma 12, $\delta = \sqrt{\frac{2(1+\frac{||\mathcal{X}||_2}{n\lambda})\log(2d/\rho)}{s}}$, which under the assumption $||\mathcal{X}||_2 = O(n\lambda)$ gives $\frac{1}{(1-\delta)}^2 = O(1 + \frac{\log(2d/\rho)}{s})^2$. Substituting in the value of $\sigma_1, \sigma_2$ gives:

$$\frac{||X||_2^4 n^2}{s^4 \epsilon^2}||\mathcal{W}||_2^2 \cdot \frac{d}{\lambda}(1 + \frac{\log(2d/\rho)}{s}) \; + \; \frac{l||\mathcal{X}||_2^2||\mathcal{Y}||_2^2}{n^2\epsilon^2} \cdot \frac{d}{\lambda}(1 + \frac{\log(2d/\rho)}{s}),$$

as desired. $\qquad\square$

### 7.6 Computational Complexity

We continue the discussion of how to efficiently compute the projection step in Algorithm 2 by exploiting properties of the Kronecker product. Recall that $C = I_d \otimes \frac{1}{n}Y_T$ where $\otimes$ denotes the Kronecker product. Then if $L\Lambda V^T$ denotes the SVD of $Y^T$, standard properties of the Kronecker product imply that the spectral decomposition of $C^T C$ is:

$$C = \frac{1}{n} \cdot (I_d \otimes L)(I_d \otimes \Lambda)(I_d \otimes V^T) \implies \tag{32}$$

$$C^T C = (I_d \otimes V)(I_d \otimes \Lambda^2)(I_d \otimes V^T) \tag{33}$$

Hence we can compute $\text{SPEC}(C)$ in the time it takes to compute $\text{SVD}(Y)$, or $O(nl\min(n,l))$. Similarly, to efficiently compute the $U^T b$ term required for Lemma 2.2 Hager (2001) we can again take advantage of properties of the Kronecker product, $U^T b =$

$$(I_d \otimes V)^T 2C^T \tilde{g} = 2(I_d \otimes V^T)(I_d \otimes \frac{1}{n}Y)\tilde{g} =$$

$$2(I_d \otimes \frac{1}{n}V^T y)\tilde{g} = \text{vec}(\frac{2}{n}V^T Y \text{mat}(\tilde{g})),$$

where $\text{mat}(\tilde{g})$ is the $l \times d$ matrix with row $i$ given by elements $(l(i-1)+1, \ldots, l(i-1)+l)$ of $\tilde{g}$, and the last equality follows properties of the Kronecker product. Now $V^T Y = \Lambda U^T$ which can be computed in $O(ln)$ since $\Lambda$ is diagonal. Multiplying by $\text{mat}(\tilde{g})$ can be done in another $O(nld)$, for total complexity of $O(nl\max(\min(n,l),d))$.

---

**Algorithm 3** Input: $\lambda, \mathcal{X} \in \mathcal{X}^n \subset \mathbb{R}^{d \times n}$, $Y = [y_1, \ldots y_l] \in \mathcal{Y}^{l \times n}$, privacy params: $\epsilon, \delta$. We denote by $\mathcal{B}$ the Algorithm in Lemma 2.2 Hager (2001)

---

$\lambda - \texttt{ReuseCov}$

1: Draw $E_1 \sim N_{d(d+1)/2}(0, \sigma_1^2)$, where $\sigma_1 = \frac{1}{n}2\sqrt{2\log(2.5/\delta)}||\mathcal{X}||_2^2/\epsilon$
2: Compute $\hat{I} = (\frac{1}{n}X^T X + E_1 + \lambda I)$
3: Compute the QR decomposition $\hat{I} = QR$
4: Draw $\hat{v} = [\hat{v}_1, \ldots \hat{v}_l] \sim \texttt{GaussMech}(\epsilon/2, \delta/2, \Delta = \frac{1}{n}\sqrt{l}||\mathcal{X}||_2|\mathcal{Y}|)$
5: Compute $\text{SVD}(Y^T) = U\Lambda V^T$
6: Compute $\hat{g} = \Pi_{C(Y)}\hat{v} = \mathcal{B}(V, \Lambda, \hat{v})$
7: **for** $i = 1 \ldots l$
8: $\quad$ Solve $R\hat{w}_i = Q^T \hat{g}_i$ by back substitution.
9: **end for**
10: Return $\hat{W} = [\hat{w}_1, \ldots \hat{w}_l]$

---

### 7.7 Sketching `ReuseCov`

Theorem 5 shows that when $n > d > l$, the complexity of Algorithm 3 is $O(nd^2)$ or the cost of forming the covariance matrix. In this section we show how sub-sampling $s < n$ points can improve this to $O(sd^2)$ by giving an analysis of sub-sampled SSP. The key ingredient is marrying the convergence of the sub-sampled covariance matrix to $X^T X$ with the accuracy analysis of SSP we saw in Section 4.

Our algorithm is based on the observation that if we sub-sample $S \subset [n], |S| = s$ points without replacement then:

- The cost of computing the covariance matrix $\Sigma_S = \sum_{k \in S} x_k x_k^T$ is $O(sd^2)$

- By the "secrecy of the sub-sample" principle Dwork & Roth (2014), our privacy cost for estimating $\Sigma_S$ is scaled down by a factor of $s/n$

- With high probability for sufficiently large $s$, $\Sigma_S \to \Sigma$ by a matrix-Chernoff bound for sampling without replacement Tropp (2010)

---

**Algorithm 4** Input: $s, \lambda, \mathcal{X} \in \mathcal{X}^n \subset \mathbb{R}^{d \times n}$, $Y = [y_1, \ldots y_l] \in \mathcal{Y}^{l \times n}$, privacy params: $\epsilon, \delta$

$\lambda - \texttt{SubSampReuseCov}$

1: Sub-sample $s$ points without replacement from $\mathcal{X}$, we denote the sub-sampled design matrix by $X_S$.
2: Let $(\epsilon_1, \delta_1) = (\frac{n}{s}\epsilon/2, \delta/2)$
3: Draw $E_1 \sim N_{d(d+1)/2}(0, \sigma_1^2)$, where $\sigma_1 = \frac{1}{n}2\sqrt{2\log(2.5/\delta)}||\mathcal{X}||_2^2/\epsilon_1$
4: Compute $\hat{I}_s = (\frac{1}{s}X_S^T X_S + E_1 + \lambda I)$
5: Draw $\hat{v} = [\hat{v}_1, \ldots \hat{v}_l] \sim \texttt{GaussMech}(\epsilon/2, \delta/2, \Delta = \frac{1}{n}\sqrt{l}||\mathcal{X}||_2|\mathcal{Y}|)$
6: **for** $i = 1 \ldots l$
7:     Set $\hat{w}_i = \hat{I}_s^{-1}\hat{v}_i$
8: **end for**
9: Return $\hat{W} = [\hat{w}_1, \ldots \hat{w}_l]$

---

**Theorem 6.** *With $\mathcal{M} = \texttt{GaussMech}(\epsilon/2, \delta/2, \Delta = \frac{1}{n}\sqrt{l}||\mathcal{X}||_2|\mathcal{Y}|)$, Algorithm 4 is an $(\alpha, \rho, O(\epsilon), \delta)$ solution to the PRIMO problem with $\alpha^2 =$*

$$O(||\hat{w}||_2^2||\mathcal{X}||_2^2|\mathcal{Y}|^2(\frac{\log(2d/\rho)}{s}) + \frac{d}{\lambda}(1 + \frac{\log(2d/\rho)}{s}) \cdot \tag{34}$$
$$(\frac{||X||_2^4 n^2}{s^4 \epsilon^2}||\mathcal{W}||_2^2 + \frac{l||\mathcal{X}||_2^2|\mathcal{Y}|_2^2}{n^2 \epsilon^2}))$$

**Proof Sketch.** Our analysis will hinge on the case where $l = 1$ e.g. that of standard private linear regression, which we will extend to the PRIMO case by our choice of $\epsilon$ as in the proof of Theorem 1. Now let:

- $w_{i*} = (X^T X)^{-1} X^T Y$ the least squares estimator as before

- $w_{i*}^\lambda = (\frac{1}{n}X^T X + \lambda I)^{-1}\frac{1}{n}X^T Y$ the ridge regression estimator

- $w_s = (\frac{1}{s}X_S^T X_S + \lambda I)^{-1}(\frac{1}{n}X^T Y)$ the sub-sampled least squares estimator

- $\tilde{w}_s = (\frac{1}{s}X_S^T X_S + E_1 + \lambda I)^{-1}(\frac{1}{n}X^T Y + E_2)$ our differentially private estimate of $w_s$

We note that $0 \le f(w_{i*}^\lambda) - f(w_{i*}) \le \lambda\left(||w_{i*}||_2^2 - ||w_{i*}^\lambda||_2^2\right) \le \lambda||\mathcal{W}||_2^2$. Then by the Lemma 2 and Cauchy-Schwartz with respect to the norm $||\cdot||_{\frac{X^T X}{n}}$:

$$|f(\tilde{w}_s) - f(w_{i*})| = ||\tilde{w}_s - w_{i*}||_{\frac{X^T X}{n}}^2 \le$$
$$3||w_{i*} - w_{i*}^\lambda||_{\frac{X^T X}{n}}^2 + 3||w_{i*}^\lambda - w_s||_{\frac{X^T X}{n}}^2 + 3||w_s - \tilde{w}_s||_{\frac{X^T X}{n}}^2 \le$$
$$3\lambda||\mathcal{W}||^2 + \underbrace{3||w_s - w_{i*}^\lambda||_{\frac{X^T X}{n}}^2}_{\text{Matrix Chernoff}} + \underbrace{3||\tilde{w}_s - w_s||_{\frac{X^T X}{n}}^2}_{\text{SSP analysis+Matrix Chernoff}} \tag{35}$$

So it suffices to bound each term with high probability. The second term, $||\tilde{w}_s - w_s||_{X^T X + \lambda I}$ can be bounded using the same arguments as in Theorem 1, with small differences due to scaling. Crucially though, as we need to bound this in the norm induced by $X^T X$ rather than $X_S^T X_S$, we will need to utilize the convergence of $X_S^T X_S \to X^T X$ via Matrix-Chernoff bounds.

**Lemma 13.** *Under the assumption $||\mathcal{X}||_2 = O(n\lambda)$, then w.p. $1 - \rho/2$:*

$$||w_s - \tilde{w}_s||_{\frac{X^T X}{n}} = O(\frac{||X||_2^4}{n^2 \epsilon^2}||\mathcal{W}||_2^2 \cdot \frac{d}{\lambda}\frac{\log(2d/\rho)}{s} + \frac{l||\mathcal{X}||_2^2||\mathcal{Y}||_2^2}{n^2 \epsilon^2} \cdot \frac{d}{\lambda}\frac{\log(2d/\rho)}{s}) \tag{36}$$

Bounding the first term can be reduced to bounding $||I - (\frac{1}{n}X^T X)^{-1/2}(\frac{1}{s}X_S^T X_S)(\frac{1}{n}X^T X)^{-1/2}||_2$ which follows more directly via the Matrix-Chernoff bound for sub-sampling without replacement:

**Lemma 14.** *Under the assumption $||\mathcal{X}||_2 = O(n\lambda)$, then w.p. $1 - \rho/2$:*

$$||w_{i*}^\lambda - w_s||_{\frac{X^T X}{n}} = O\left(\frac{||\mathcal{X}||_2|\mathcal{Y}|\log(2d/\rho)}{\lambda s}\right)$$

Substituting into Equation 36 and minimizing over $\lambda$ gives the desired result.

### 7.8 Additional Experimental Results

**Realizability of outcomes has a minimal effect.**

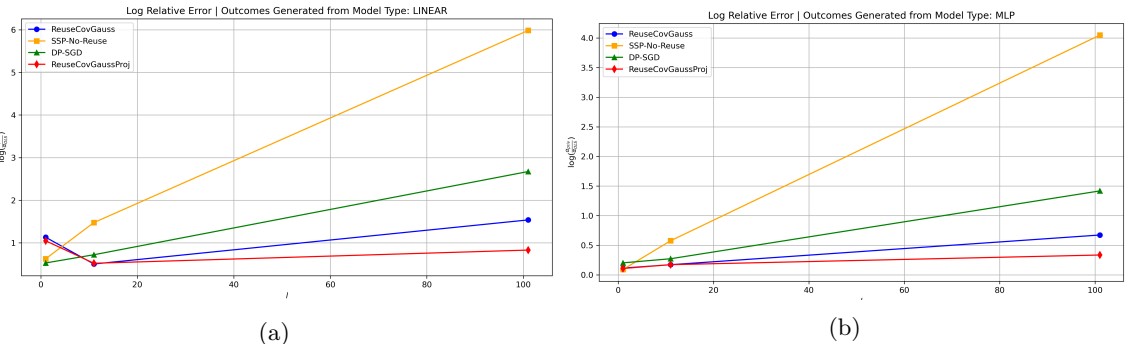

(a)  (b)

Figure 5: Both (a) and (b) show the log of the ratio of the squared loss of the private estimator to the square loss of the OLS estimator, but in (a) the outcomes are synthetically generated from a linear model, whereas in (b) the outcomes are generated from a 2-layer neural network, to test how the algorithms perform when the outcomes are not generated by a linear model. All values are averaged over 5 iterations, over $l = (1, 11, 101)$ with $(\epsilon, \delta) = (5, \frac{1}{n^2})$ and $n = 5000, d = 25$ on a synthetic dataset with Gaussian features.

**DP-SGD outperforms `ReuseCov` for $l < \frac{n}{\sqrt{d}}$.** We construct our dataset by sub-sampling $n = 5000$ images of dimension $d = 784$ from MNIST Deng (2012), and generating $l = 1, 11, 101$ outcomes from a noisy linear model with unit norm. We implement DP-SGD using the Opacus Yousefpour et al. (2021) library from Meta, and do privacy accounting across regressions by composing in zCDP  and then converting back to $(\epsilon, \delta) - $DP. Figure 6 shows that `DP-SGD` outperforms our `ReuseCov` algorithms when $l \in (1, 11, 101)$, although, in agreement with the theory, this gap shrinks as $l$ increases.

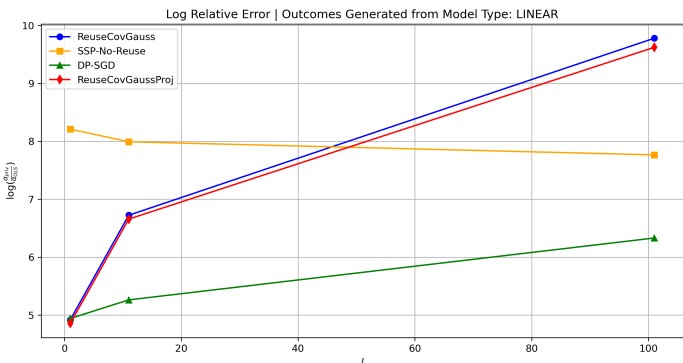

Figure 6: We construct our MNIST dataset by sampling $n = 5000$ random training images ($d = 784$), and then generating synthetic labels from a random linear model. All values are averaged over 5 iterations, over $l = (1, 11, 101)$ with $(\epsilon, \delta) = (5, \frac{1}{n^2})$.

