# OpenReview forum: "PRIMO: Private Regression in Multiple Outcomes"
_TMLR — Accepted by TMLR_

### Review · Reviewer_gBuC · 2024-10-04

**Summary Of Contributions:**

This paper investigates differentially private linear regression in multi-target settings, where the label $Y\in\mathcal{R}^{l}$ is a vector rather than a scalar. All targets share the same weight matrix.

**Audience:**

Yes

**Claims And Evidence:**

Yes

**Requested Changes:**

Please see weaknesses.

**Strengths And Weaknesses:**

The primary focus of the paper is on objective perturbation, with the authors proving utility bounds in three settings: full-DP, feature-DP, and label-DP. For the full-DP setting, the authors propose the ReuseCovGauss method, which reuses the noisy covariance matrix $X^{T}X$ for all targets, where $X\in \mathcal{R}^{n\times d}$ is the feature matrix. This approach yields improved bounds compared to the naive baseline that applies objective perturbation separately for each target. In the feature-DP setting, the authors provide a bound that is independent of the number of targets$l$. The authors conduct experiments on a genome dataset to support their theoretical results.

Overall, the paper addresses a relevant and interesting problem in differentially private optimization. I have the following suggestions for improvement:

1. The authors use privatized gradient descent as a baseline for their theoretical comparisons. It would be good to also include this method as a baseline in the experimental results.

2. A table summarizing the conditions under which the bounds derived in this paper outperform or compare to the baseline methods would help clarify the paper's contributions.

3. There is a missing bracket in Eq. (16).

---

> ### Author Response · Authors · 2024-10-18
> **Additional Private Baselines + Summary Table**
>
> We thank the reviewer for their suggestions to implement DP-SGD as a private baseline, as well as adding a more clear summary table that shows the advantage of our approach over existing results.  We want to highlight that in Figure $2$ we do show a column $\text{cost}(l)$ which is the provable improvement in MSE that results from our method relative to the $\textit{best possible}$ MSE that can result from repeating a single estimation procedure $l$ times. Is there another way to summarize that you think would be more helpful? One approach could be to take a specific method like DP-SGD and compare.
>
> We will follow up when we have added additional private baselines to the experimental results.

---

> > ### Author Response · Authors · 2024-11-21
> > **New Experimental Results Requested**
> >
> > Dear reviewer, thank you for your patience as we added the requested results to the paper. We summarize our response to your questions below:
> >
> > 1. We have added DP-SGD as a baseline for comparison, see the green curve in Figure 6 (a) and 6(b). As expected, at very small values of $l$, DP-SGD is a state of the art method and outperforms SSP-variants, however, at even modestly large $l > 101$ both of our SSP-variants (ReuseCovGauss and ReuseCovProj) outperform DP-SGD significantly (note the y-axis is on the log scale here). This is consistent with our theory, as the error of DP-SGD scales like $\sqrt{l}$. We are currently running simulations for larger values of $l$, although these should only improve the performance of our methods relative to DP-SGD and so likely would only be included in the Appendix.
> > 2. We direct the reviewer to Figure 2 (see our response above) and welcome any additional feedback on this table.
> > 3. Addressed thank you.

---

### Review · Reviewer_Jq7n · 2024-10-13

**Summary Of Contributions:**

This submission considers differentially private (DP) linear regression where each individual contributes a tuple of $d$ attributes and $\ell > 1$ outcomes. The naive baseline would be to run a DP optimizer $\ell$ times, but this incurs an overhead of $\sqrt{\ell}$ in the error as measured in squared-Froebenius norm. The authors begin with the observation that the $\ell$ executions can re-use a noisy covariance estimate, reducing error in a ``small $\ell$'' regime. For other values of $\ell$, the authors show reductions in error are possible when it is permitted to remove protections from either features or labels. Experiments indicate that these relaxations bring the error close to that of the non-DP OLS estimator.

**Audience:**

Yes

**Claims And Evidence:**

Yes

**Requested Changes:**

As per the *Weaknesses*, extend Table 1 to include comparison with the naive baseline. Also, if time permits, experiments with other datasets where label or feature DP is more applicable than the genomics case.

**Strengths And Weaknesses:**

*Strengths*
Sections 1 and 3 are easy to read. I particularly liked the clarity of the paragraph beginning "We answer this question..."

The techniques are straightforward but yield clear improvements over the naive baseline.

I anticipate there are at least a few applications where PRIMO applies and label DP or feature DP is acceptible.

*Weaknesses*
The submission is missing comparison with the naive baseline of using the optimal single-outcome mechanism $\ell$ times. In particular, I would like to see how it fares in each of the $\ell$ regimes that Table 1 decomposes, both asymptotically and experimentally. I also think this is a more reasonable benchmark for comparison than the lower bound by Bassily et al. (2014), since that result is limited to full DP and is "not necessarily tight for linear regression" as the authors admit.

In the genomics use-case that the submission focuses on, it is unclear what label and feature DP guarantee to an individual. Is it realistic to assume that the thousands of SNPs belonging to an individual are fair game to be processed without protections? On the face of it, it seems dangerous. The authors do acknowledge this in the Limitations section, but the experimental section still heavily focuses on algorithms for genomics that do not satisfy full DP; this raises the question of why they did not consider other datasets for evaluation.

A limitation not discussed in the submission is the fact that the labels in the experiments were generated according to a random linear model. While this is a reasonable "sanity-check"---if there were no regime where an algorithm performed well on such data, it would clearly not be a good linear regression algorithm---I would caution that the experimental results are not as strong as if they were obtained by operating on real labels. This is more motivation to move to other datasets where labels are easier to come by than the genomics case.

*Questions*
In the experiments, was ReuseCovProjX used or ReuseCovProjY? The "Datasets" paragraph of Experiments indicates the latter but the "Experimental Details" indicate the former. And why not test both?

What happens when we restrict the privacy definition? Specifically, if we are limited to Renyi, concentrated, or pure DP?

---

> ### Author Response · Authors · 2024-10-18
> **Additional Datasets and Baseline Comparisons**
>
> Dear Reviewer, thank you for your very constructive review. The main suggestions are to add experiments on additional datasets that might be better justified from the perspective of label-DP or feature-DP than in the genomic setting, and datasets where the outcomes are not synthetically generated from a linear model. Although prior work in genomics has focused on label-DP https://pubmed.ncbi.nlm.nih.gov/27453444/ (in this case called phenotypic privacy), we agree that the paper could benefit from other motivating use-cases. The reviewer also suggests comparison with the baseline of just running SSP $l$-times and applying advanced composition, which we plan to incorporate.
>
> We will follow up in a few days once the additional results have been incorporated.

---

> ### Comment · Reviewer_Jq7n · 2024-11-02
> **Deadline notification**
>
> hello authors,
>
> Can you give an update on the changes you planned to make? We can be more confident in our scores if we're sure everything has been addressed. Official recommendations are due Nov 11.
>
> Thank you.

---

> > ### Author Response · Authors · 2024-11-11
> > **Response**
> >
> > Dear reviewer, we have been given an extension until Nov 15th to obtain preliminary results but the main changes are:
> > - additional dataset where the outcomes are not "realizable" e.g. generated from a linear model -- we are trying an MLP
> > - results comparing to objective perturbation / DP-SGD which both fail miserably because the error sales with $\sqrt{l}$ the number of outcomes
> > - reorganization of a few parts of the paper (section 3 for example) to more clearly separate results from methods.
> > Thank you and we'll be in touch soon.

---

> ### Author Response · Authors · 2024-11-21
> **Summary of Changes**
>
> Dear Reviewer,
> Thank you for your patience. Please direct your attention to Figures 6(a), 6(b), and 7, in the updated draft.
> To address each of your requested changes:
>
> **1. Comparisons to the naive baseline of running $l$ times**
> In Figure 6(a) we have added two "naive" private baselines -- DP-SGD, the state of the art private optimization method, and SSP without reusing the covariates. At $l = 1$ (the non-PRIMO setting) as expected DP-SGD outperforms the other methods, but even by $l = 100$ both naive methods lag significantly behind the PRIMO algorithms. Recall that the Y-axis in this plot is on the log-scale and so the improvement over the naive baseline is already quite significant.
>
> **2. Additional Datasets for Comparison**
> Figures 6(a) and 6(b) add two synthetic datasets, where the features are generated from a random gaussian, and the outcomes are synthetically generated. Figure 7 adds the MNIST dataset with synthetically generated outcomes. In Figure 7 $d$ is larger relative to $l$ and so we are in the regime (row 2 of Figure $2$) where we expect no improvement from our methods; indeed in this case we see that DP-SGD is the best performing method.
>
> **3. Assumption of a random linear model**
> Although we still rely on synthetically generated outcomes in order to vary $l$ over large values, in Figure 6(b) we add experiments where the outcomes are NOT generated from a linear model as you suggest -- instead we generate them from a 2 layer MLP. Our main findings remain unchanged in this setting, because the relative ordering of the different methods remains the same -- our PRIMO variants still outperform DP-SGD and SSP at $l$ as small as $11$. One difference between the two settings is that in general, all of the methods perform worse (relative to the non-private OLS estimator)  when the outcomes are generated from MLP rather than when they are generated by a linear model. Explaining this theoretically could be an interesting avenue for future work.
>
> **4. Experiments with $Proj_X$ vs $Proj_Y$**
> Our experiments focus on $Proj_X$ as label differential privacy is the more widely studied relaxation of DP relative to feature-DP. That being said, we can easily add experiments with $Proj_Y$ to the final version.

---

### Review · Reviewer_qHFs · 2024-10-14

**Summary Of Contributions:**

This paper studies differentially private linear regression problems, which aim to protect the data privacy of the linear regression problem (with vector labels) when releasing the final model. The paper proposes a series of privacy-preserving algorithms by applying the Gaussian mechanism to different parts of the regression problems. The authors further provide the theoretical error bounds, i.e., the privacy-utility trade-off of these algorithms under different settings.

**Audience:**

No

**Broader Impact Concerns:**

I didn't see Broader Impact Concerns in this paper.

**Claims And Evidence:**

Yes

**Requested Changes:**

Unifies the notations:
1. The optimal/private solutions $W$ or $w$ have multiple definitions, which might be confusing.
2. The notation on the top two lines on Page 4 is confusing; please provide more explanation.

Figure 2 is a table.

Please improve the English usage and check the grammar. I suggest a complete rewrite of the paper.

Typos:
1. Improper citation format, e.g., missing parenthesis on page 1 citing Dwork & Roth (2014).
2. eq(3), missing ")", $w_{private}, w_{opt}$ not defined.
3. Input of algorithm 1 $\mathcal{X}$ should be $X$.
4. What is normregw in eq (22)?

**Strengths And Weaknesses:**

Strength:
1. The paper provides a strong theoretical analysis of DP multiple Linear Regressing (PRIMO) under different settings, including full DP, feature DP, and label DP with different ranges of data set sizes, feature dimensions, and label numbers.
2. The dependency on label size $l$ is interesting.

Weakness:
1. The paper is poorly organized and badly written. For example, Section 3 mixes the summary of the results with the description of the algorithms and intuitions; the authors cite equations from another paper without clear explanations (eq.3, 5). Lemmas and Theorems, and their proofs are directly listed in the paper without clean organization. Lemmas from other papers are not clearly cited, e.g., Lemma 2.
2. Most results directly apply existing methods and analyses, e.g., Lemma 2, 3, 4, 5, 6, and 7 are either citing results from other paper or well-known inequalities. The novelty of the paper is not significant.

---

> ### Author Response · Authors · 2024-10-18
> **Organization and Grammar**
>
> Dear Reviewer, thank you for your comments. If we are reading your review correctly, you aren't requesting any changes to the results or analysis, but you want us to focus on the grammar and structure of the paper, in addition to fixing some small typos. You highlight Section 3 as being badly organized, although we will remark that one of the other reviewers specifically mentions Section 3 as being easy to read. In addition to taking a pass for grammar, and separating out a summary of results from algorithmic descriptions, are there other changes to the high-level flow that you want us to consider making?
>
> Just to remind you the paper is currently organized as follows:
>
> 1. Intro: Introducing the novel private regression with multiple outcomes (PRIMO) problem
> 2. Preliminaries: Defining label and feature level DP, the PRIMO problem, the Gaussian Mechanism
> * 2.1: Summarizing existing work on private linear regression with a single outcome
> 3. Summarizing main results and describing the intuition behind ReuseCovProj
> 4. ReuseCovGauss: Analysis of the basic Gaussian Mechanism based algorithm
> 5. Improved Algorithms for Large L: Analysis of ReuseCovGauss with the random Projection
> * 5.1 Feature DP & 5.2 Label DP -- theoretical results analyzing how the projection reduces error significantly in certain parameter regimes.
> 6. A discussion of the computational complexity of these baseline methods
> * 6.1 Baseline computational complexity without any sketching
> * 6.2 Faster algorithms that use random sketching (including a theoretical analysis of the error introduced by sketching)
> 7. Empirical Results on Genomic Data showing our methods outperform the baselines

---

> > ### Comment · Reviewer_qHFs · 2024-10-31
> > **Request for updating the writing**
> >
> > Dear authors, Thank you for replying. My major concern is the writing/grammar and format of the paper. The technical results are rigorously stated and proven. Other than that, I don't have any specific concerns.

---

### Comment · Action_Editor_fHx8 · 2024-10-16
**Discussion phase has begun**

Authors, all reviews for this paper have been received. You are encouraged to engage with the reviewers during this phase. Please submit any responses in the next 2 weeks.

---

> ### Comment · Action_Editor_fHx8 · 2024-11-16
> **Revisions will be submitted early next week**
>
> Dear reviewers,
>
> The authors have informed me that intend on submitting their revisions early next week. Please wait until then to submit your final recommendations.

---

### Author Response · Authors · 2024-11-22
**Summary**

Dear Reviewers,
We have responded to all requested changes below, and have run several new experiments (2 new private baselines, 2 new datsets) to address gaps in the existing work. In addition if the paper is accepted we commit to:
1) implementing all grammatical and cosmetic changes pointed out by the reviewers
2) adding details about the new experiments and plots into the experimental details section
3) we have ongoing simulations for the new baselines with larger values of l that are quite time consuming to run

Please let us know if further changes would be appreciated.

---

### Decision · Action_Editor_fHx8 · 2024-12-11

**Recommendation:** Accept as is

**Comment:**

Reviewer comments have been sufficiently addressed during revision period.

**Audience:**

The reviewers all agreed that there is sufficient interest in this paper among the TMLR audience.

**Claims And Evidence:**

Reviewers agreed that the techniques showed clear improvement over prior work and would be of interest to the community.